# Multi-omics signatures of the human early life exposome

Environmental exposures during early life play a critical role in life-course health, yet the molecular phenotypes underlying environmental effects on health are poorly understood. In the Human Early Life Exposome (HELIX) project, a multi-centre cohort of 1301 mother-child pairs, we associate individual exposomes consisting of >100 chemical, outdoor, social and lifestyle exposures assessed in pregnancy and childhood, with multi-omics profiles (methylome, transcriptome, proteins and metabolites) in childhood. We identify 1170 associations, 249 in pregnancy and 921 in childhood, which reveal potential biological responses and sources of exposure. Pregnancy exposures, including maternal smoking, cadmium and molybdenum, are predominantly associated with child DNA methylation changes. In contrast, childhood exposures are associated with features across all omics layers, most frequently the serum metabolome, revealing signatures for diet, toxic chemical compounds, essential trace elements, and weather conditions, among others. Our comprehensive and unique resource of all associations (https://helixomics.isglobal.org/) will serve to guide future investigation into the biological imprints of the early life exposome.

A large proportion of environmental risk factors remains unknown or poorly defined, although the environmental contribution to disease risk is estimated to be 70–90%[1,2]. More than a decade ago, the term "exposome" was coined to encompass all environmental factors (i.e. non-genetic factors) to which humans are exposed throughout the life course[3]. Historically, environmental health studies focused almost exclusively on single exposure factors such as air pollution, lead, or pesticides. The central tenet of the exposome concept is a call for a holistic and systematic approach to assessing the impacts of environment on health. Moreover, the exposome includes not only external exposures, but also the internal biological responses to these exposures through the interrogation of high-dimensional molecular data[3–6].

Of particular interest is the early detection of physiological changes at the molecular level related to environmental exposures before the manifestation of clinical symptoms in healthy populations. Such information may support the biological plausibility of environment-health associations in population studies, help to understand toxicological mechanisms or elucidate how multiple exposures may be grouped based on their common influence on

biological pathways (e.g. inflammation) or their source of exposure (e.g. diet). It can also help to identify exposure biomarkers to predict current and past exposures. Integrative personal omics profiling studies, gathering high-throughput data on multiple molecular layers, have demonstrated that personal molecular profiles may be particularly useful to assess disease risk, detect early preclinical conditions and initiate preventive strategies[7–9].

Foetal and childhood development has life-long consequences and is critical for many chronic diseases including obesity, cardiometabolic diseases[10–12], attention-deficit and hyperactivity disorders (ADHD)[13] and lung function[14]. Therefore, early life is a particularly important period to study the early biological triggers of disease: exposures during these developmentally vulnerable periods may have pronounced effects at the molecular level that may remain clinically undetectable until adulthood.

The molecular mechanisms through which early-life environmental exposures may impact birth outcomes and long-term health in humans have primarily been studied through the lens of epigenetics. It is thought that the epigenome orchestrates cellular responses to

e-mail: martine.vrijheid@isglobal.org

environmental perturbations and provides cell memory and plasticity[15]. Among all epigenetic marks, DNA methylation is the most studied in epidemiological settings; and among all exposures, tobacco smoke is the most investigated[16–19]. To a lesser extent, other diverse exposures, from metals and air pollution to socio-economic factors, have been linked to differential methylation and are catalogued in public databases[16] (http://www.ewascatalog.org/). Although epigenetic marks regulate gene transcription and thus the proteome, the relationships between these and the exposome are less studied[17]. The metabolome, which can reflect physiological responses and microbiome activity, as well as the direct internalization of exposures, has received particular attention in exposome research[5,18–20]. However, there is a clear lack of large-scale studies that evaluate multi-omics signatures of a wide range of environmental exposures.

In this work, we aimed to associate the personal early life exposome, measured in 1301 mother–child pairs of the Human Early Life Exposome (HELIX) project, with deep molecular phenotype data assessed in childhood and defined by the blood methylome and transcriptome, plasma proteins, and serum and urinary metabolites[21]. By systematically documenting all associations between the exposome and the molecular phenotypes, we provide a unique resource (https://helixomics.isglobal.org/) for the identification of novel exposure biomarkers and early biological effects during developmentally vulnerable life periods.

## Results

### Building the early life exposome and the multi-omics phenotypes in HELIX children

We assessed the early life exposome in 1301 mother–child pairs from the HELIX project, a multi-centre longitudinal population-based cohort study in 6 locations in Europe (Spain, UK, France, Lithuania, Norway and Greece) (Fig. 1 and Supplementary Data 1A)[21]. We measured 91 environmental exposures in pregnancy and 116 in childhood, when children were between 6-11 years old. Exposures covered 19 families: meteorological factors, natural spaces, indoor and outdoor air pollution, built environment, road traffic, noise, water disinfection by-products, tobacco smoking, lifestyle factors (diet, physical activity), social and economic capital, essential minerals and chemical pollutants (non-essential metals, organochlorines, organophosphate pesticides, polybrominated diphenylethers, perfluoroalkyl substances, phenols and phthalates) (Fig. 1). Exposure levels in the HELIX cohorts are described further elsewhere[22–24]. Correlation patterns among exposure variables adjusted for cohort are shown in Supplementary Data 1B (pregnancy), 1C (childhood) and 1D (for the same exposure variable among the two periods). Exposure assessment tools included mass spectrometry-based measurement of biomarkers of chemical exposure in urine and blood, exposure monitors, remote sensing and geospatial methods, and questionnaire-based interviews.

For these same children, aged between 6 and 11 years, we performed in-depth multi-omics molecular phenotyping, including measurement of blood DNA methylation (450K, Illumina), blood gene expression (HTA v2.0, Affymetrix), blood miRNA expression (SurePrint Human miRNA rel 21, Agilent), plasma proteins (3 Luminex multiplex assays), serum metabolites (targeted LC-MS/MS metabolomic assay, Biocrates AbsoluteIDQ p180 kit), and urinary metabolites ($^1$H nuclear magnetic resonance (NMR) spectroscopy) (Fig. 1 and Supplementary Data 1E). While blood DNA methylation and transcriptomics were measured genome-wide with 386,518 CpGs, 58,254 transcript clusters (TCs) and 1117 miRNAs; the other omics followed a semi-targeted or targeted approach. Plasma proteins included a total of 36 cytokines, apolipoproteins and adipokines (Supplementary Data 1F)[25]. The serum metabolites ($N = 177$) included amino acids, biogenic amines, acylcarnitines, glycerophospholipids, sphingolipids and sum of hexoses, covering a wide range of analytes and metabolic pathways in one

targeted assay (Supplementary Data 1G)[26]. Urine metabolites ($N = 44$) mainly included amino acids, organic acids, nicotinamides, amines and gut microbial-derived phenols (Supplementary Data 1H)[26]. Around 91% of the children had molecular data from at least 4 of the omics platforms. Detailed information on the HELIX participants, exposure assessment and omics measurements can be found in Supplementary Information.

### Results of the exposome-omics-wide association study (ExWAS)

We first systematically tested the association between each exposure variable and each molecular feature, successively and independently, through an ExWAS, using an analogous statistical approach to that of Genome-Wide Association Studies (GWAS) (Fig. 1 and Supplementary Information). Overall, we tested >30 M exposure-omics associations (>0.3 M molecular features * ~100 exposures * 2 exposure periods) through linear regression models adjusted for the same set of confounders: cohort, child's age, sex, z-score body mass index (zBMI), ancestry, maternal education and omics specific covariates. Results of all these associations can be viewed in the web catalogue: https://helixomics.isglobal.org/ (for genome-wide omics platforms, only results with p values <0.01 are included).

To identify statistically significant exposure-omics associations, correction for multiple comparisons was applied for each exposure within each omics dataset. For this, we considered significant associations the ones with p values below a False Discovery Rate (FDR) of 0.05 for genome-wide omics, and below a modified version of the Bonferroni cut-off for the proteins and metabolites (which consists in dividing the nominal p value by the effective number of tests (ENT) determined from the correlation structure of the omics dataset (Supplementary Data 1I and Supplementary Information). With these criteria, 1170 exposure-omics associations were statistically significant. Associations between the pregnancy exposome and molecular phenotypes totalled 249, including 52 unique exposures and 209 unique molecular features, while the 921 associations with the childhood exposome corresponded to 84 unique exposures and 454 unique molecular features. All 1170 statistically significant associations are shown in Supplementary Data 2.

Miami plots display exposure-omics associations by family of exposure and molecular layer (Fig. 2A1, B1). The pregnancy exposome was predominantly associated with child DNA methylation (70% of the associations observed) (Fig. 2A2); in contrast, the childhood exposome was associated with all molecular layers, with the serum metabolome showing the highest number of associations (43% of the associations observed) (Fig. 2B2). Pregnancy exposures within the most associations included molybdenum (Mo), cadmium (Cd), cotinine (biomarker of tobacco exposure) and maternal smoking (questionnaire data) (Fig. 2A3). Childhood exposures with the most associations included copper (Cu), organochlorine compounds (PCB 118), and perfluroalkyl substances (PFOS), caesium (Cs) and humidity (Fig. 2B3). Other exposures such as outdoor air pollution, built environment, road traffic, and noise, showed few associations. Among 83 exposures measured in both the pregnancy and childhood periods, 14 exposure-omics pairs were statistically significant in the two periods: 6 CpGs related to tobacco smoking, and several long chain fatty acids related to cotinine, hexachlorobenzene (HCB), perfluoroundecanoate (PFUnDA) and Hg (Supplementary Data 3).

**Robustness of results with respect to ancestry, child zBMI and cohort.** For the 1170 significant exposure-omics associations, we conducted several sensitivity analyses. First, HELIX consists of 1171 European ancestry children and the rest from other ancestries, with Pakistani ancestry the second most common (102 children). We repeated the ExWAS in children only of European ancestry, and did not note substantial differences in effect size (i.e. more than doubling) between the two models (Fig. 3A).

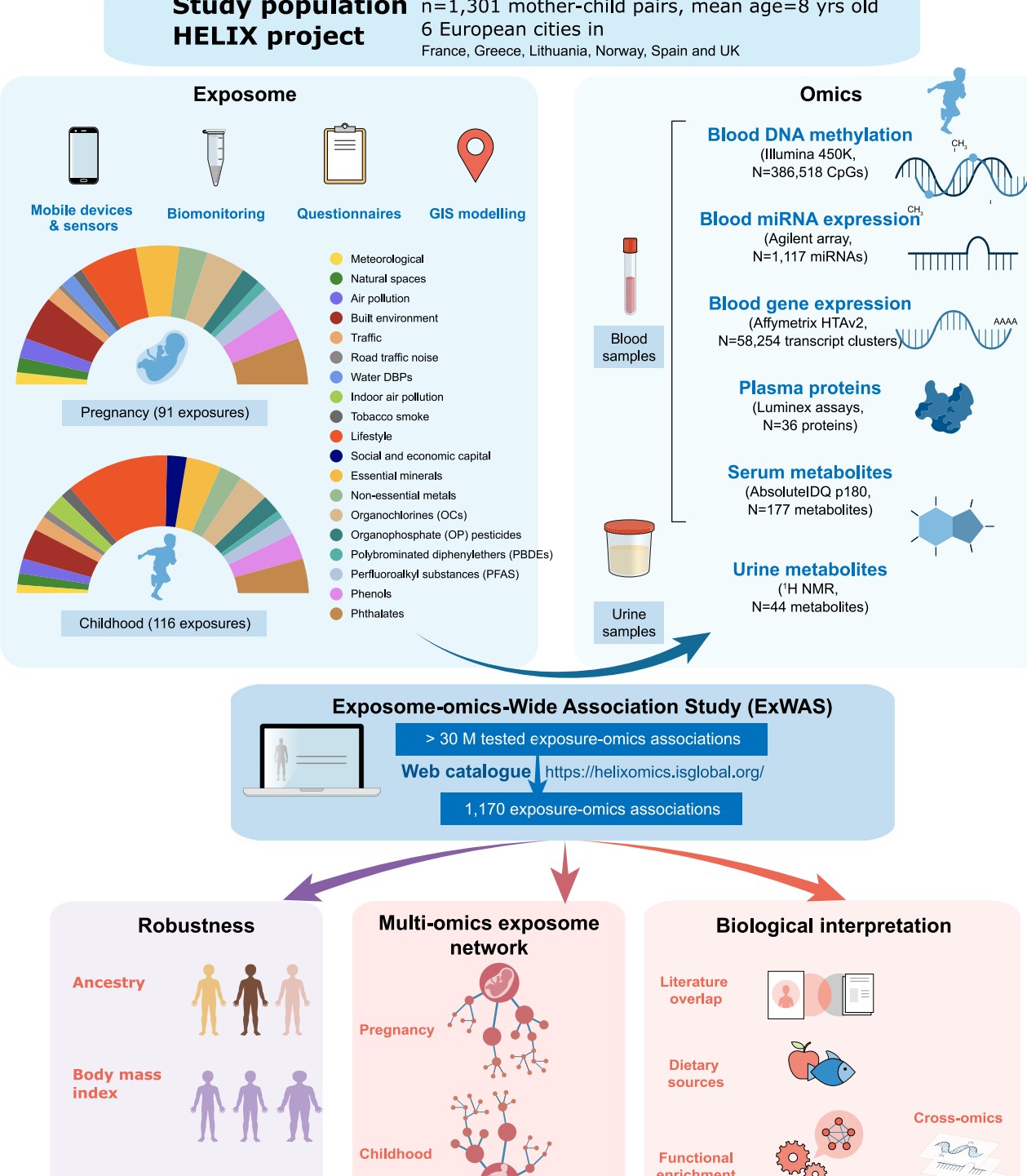

**Fig. 1 | An overview of the early-life exposome and multi-omics signature study.**
One thousand three hundred and one mother–child pairs from the HELIX project participated in the study. The early-life exposome was assessed in pregnancy and childhood through the use of different methods. The pie charts represent the proportion of exposures assessed per exposure family. Molecular traits in the child were measured using six different omics platforms using blood (blood cells, serum or plasma) or urine. Then, an Exposome-omics-Wide Association Study (ExWAS) was conducted, modelling exposure-omics one by one and adjusting for confounders. All summarized results can be found in https://helixomics.isglobal.org/. In all, 1170 exposure–omics associations passed multiple testing correction threshold. After checking the robustness of these associations to ancestry, BMI and cohort, they were visualized through multi-omics exposure networks. Finally, we did biological interpretation including overlap with the literature, identification of dietary sources, functional enrichment analyses and cross-biological matrix and cross-omics comparisons.

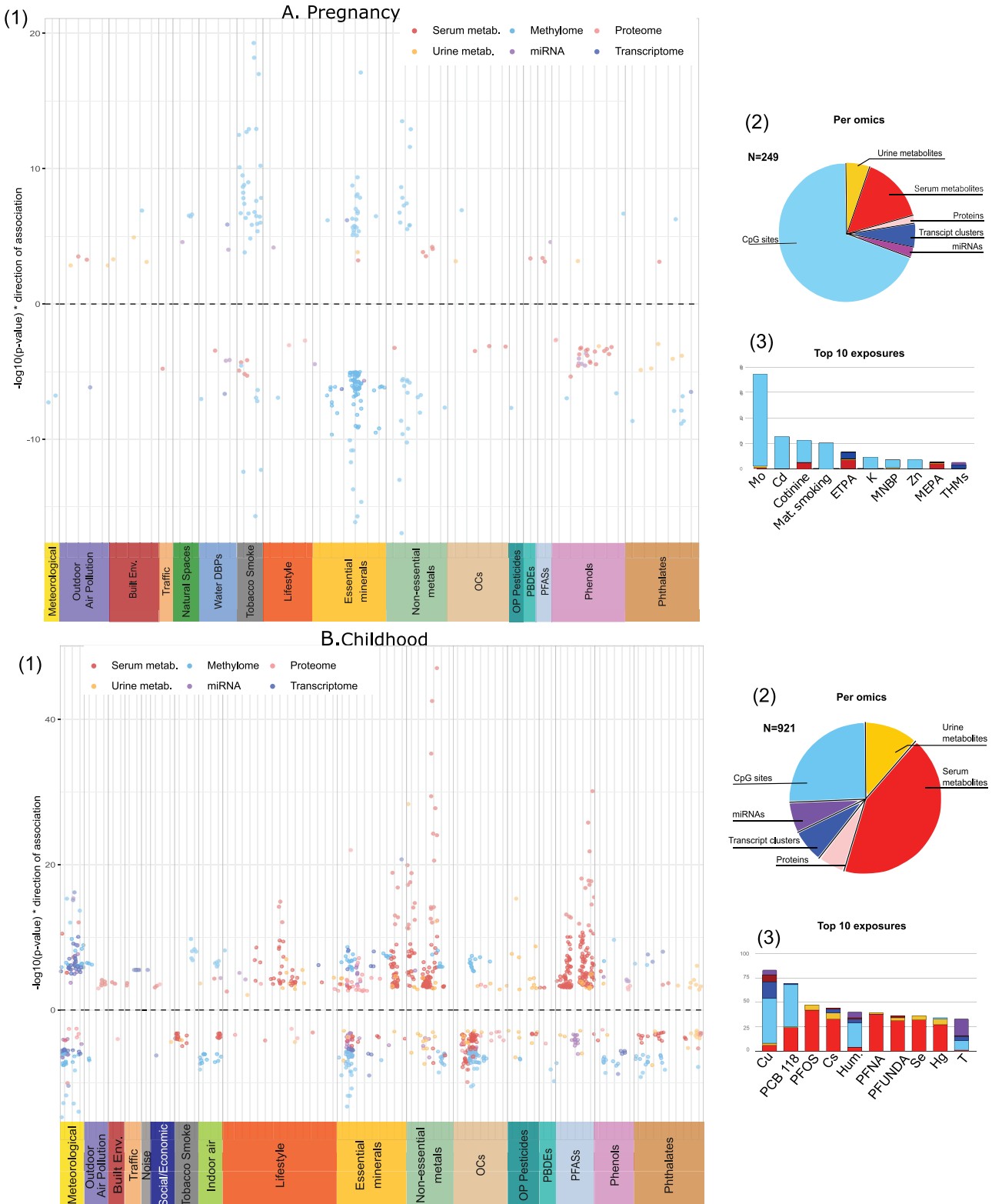

**Fig. 2 | Results of the Exposome-omics-Wide Association Study (ExWAS) for the pregnancy and childhood exposomes. A** Summary of the associations between the pregnancy exposome and multi-omics measured in 1301 children: Miami plot (1); pie charts showing the proportion of associations with the different molecular layers (2); and top 10 pregnancy exposures (3). **B** Summary of the childhood exposome-child omics associations: Miami plot (1); pie charts showing the proportion of associations with the different molecular layers (2); and top 10 childhood exposures (3). In Miami plots, each point corresponds to an exposure-omics association; the *y*-axes show the −log10 *p* values multiplied by the direction of the association (sign of the regression coefficient); and the *x*-axis groups exposures along the 19 exposure families and each vertical line represents a separate exposure with some jitter added to avoid overlapping points. In the Manhattan (dots), pie-chart and histogram, colours indicate the molecular layer.

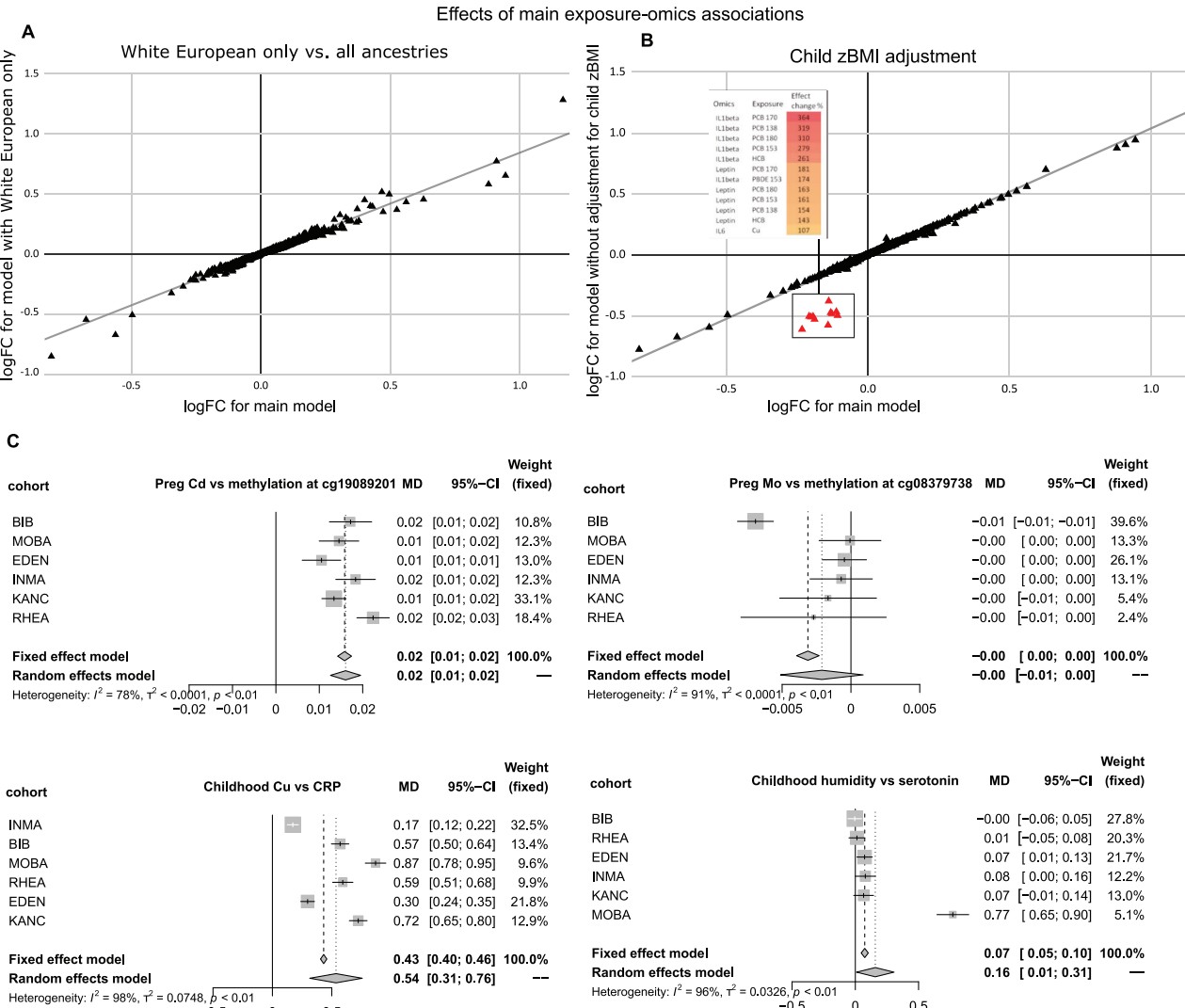

**Fig. 3 | Robustness of main exposome-omics associations.** Comparison of effect sizes of the 1170 exposome-omics associations of the main model, that includes all children (*N* = 1301) and is adjusted for child's zBMI and ancestry, vs. effect sizes of alternative models. Each triangle represents an association. The *x*-axis represents the effect size of the exposure on the omics feature in the main model, while the *y*-axis represents the effect size in the alternative model. The percentage change in effect size between models is calculated as indicated in the Supplementary Information. **A** Alternative model included all covariates but was restricted to European ancestry children (*N* = 1171). No major differences were observed. **B** Alternative model included all children and was unadjusted for child's zBMI. Exposure-omics associations with a percent change between models above 100% are coloured in red, and include proteins and child lipophilic chemical pollutants, as listed in the table. **C** Forest-plots showing the fixed- and random-effects inverse variance

weighted meta-analyses of illustrative exposure–omics associations: maternal Cd levels and child DNA methylation at CpG cg19089201 (*MYO1G* gene) (*n* = 1173), with consistent effects across cohorts; child Cu levels and child CRP levels in plasma (*n* = 1170), with consistent effects across cohorts; maternal Mo levels and child DNA methylation at CpG cg08379738 (*DENND1C* gene) (*n* = 1173), driven by one of the cohorts (BiB); humidity in childhood (1 month before sampling) and child serum serotonin levels (*n* = 1198), driven by one of the cohorts (MoBA). Each cohort is represented by a point estimate, bounded by the 95% confidence interval (CI) for the effect and the cohort weight as a grey square. The 95% CI from the fixed and random effects meta-analysis are shown as diamonds. The effect size is reported as a log2 fold change (log2FC) of the omics, or difference in methylation levels, for interquartile range (IQR) of continuous exposure variables.

Second, due to the potential influence of child adiposity both on the blood levels of lipophilic pollutants and on some molecular features, we compared the associations with and without adjustment for child zBMI, as a proxy of child adiposity. We observed that 12 associations had more than a doubling in the effect size (Fig. 3B). They included lipophilic chemicals (PCB 170, PCB 153 and PCB 180) and proteins known to be produced by the adipose tissue (IL1beta, leptin and IL6).

Third, we investigated heterogeneity across cohorts by running the 1170 exposure-omics associations by cohort. Around half of all associations presented heterogeneity values (*I²*) < 0.5, with variations by period and molecular layer (Supplementary Information–Fig. S1). Besides the *I²* statistic which might be overestimated in meta-analysis

with a small number of studies[27], we also visually inspected the forest plots. While some associations seemed to be very consistent between cohorts even with a high *I²* (e.g. maternal cadmium and methylation at CpG cg19089201), for others there was more heterogeneity with some cohorts acting as outliers (e.g. child meteorological conditions and serotonin) (Fig. 3C). The forest-plots for the 1170 exposure-omics associations are provided in Supplementary Data 11.

Finally, given the correlated nature of the exposome, we ran multi-exposure models for those omics features associated with more than one exposure, when these exposures had a correlation <0.8 and belonged to different exposure families (except individual exposures that belonged to diet, metals or parabens that we considered as separate groups). Results of these analyses are shown in

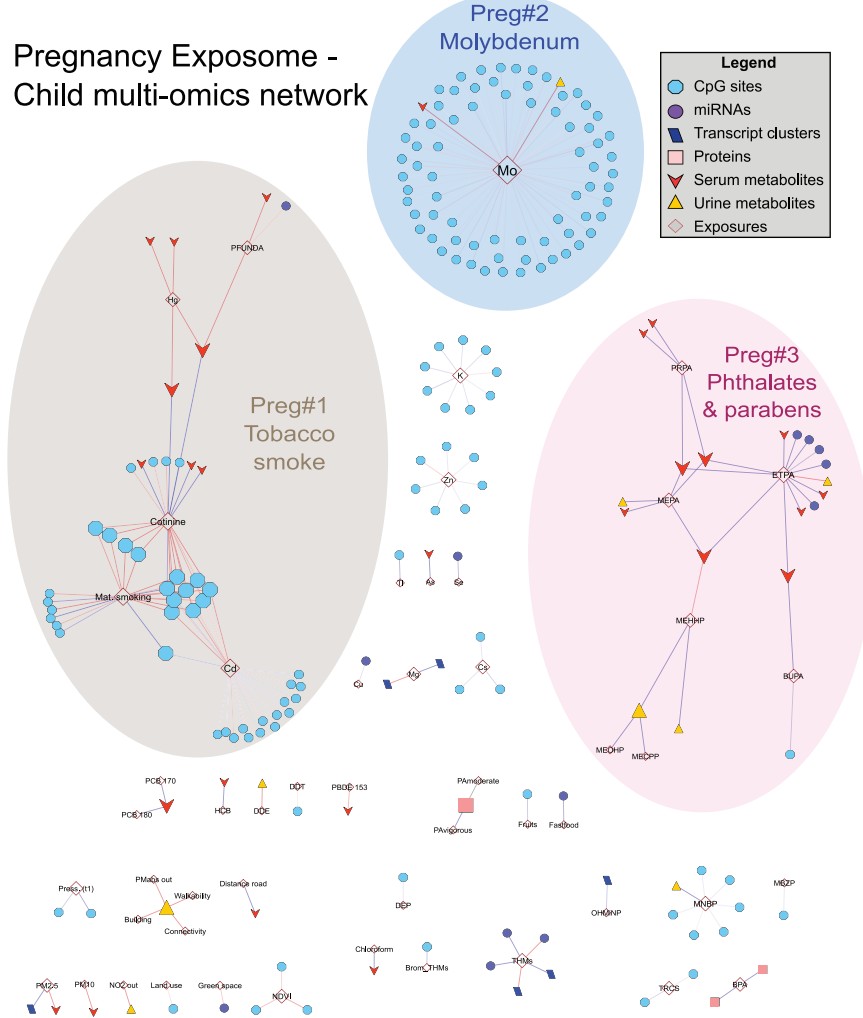

**Fig. 4 | Network map of the multi-omics signatures of the pregnancy exposome.** Network visualization of the pregnancy exposome-omics-wide association study (ExWAS). An exposure and a molecular feature were connected if their association was statistically significant (blue if positively and red if negatively). Only connected components with at least two molecular features were displayed. Nodes of the network are depicted with a different colour/shape depending if they are exposures or features of a particular molecular layer (see legend in the figure). Three main connected components were annotated, which varied greatly by their size, their number of exposures and the type of omics composing them. The summary table with the cluster characteristics are in Table 1 and a full table with the node attributes can be found in Supplementary Data 5A.

Supplementary Data 4A, B. For prenatal exposures, the strongest effect change was observed for maternal cadmium (Cd) levels, which showed a reduction of >25% of the association with the molecular trait adjusting for smoking related variables. For childhood exposures, the strongest effect changes were observed for Hg, As, Se, PFAS and dietary patterns (e.g. fish and KIDMED score), for indoor PM and parental smoking, and for BPA and meteorological variables. They are discussed below in more detail.

**Network integration of multi-omics signatures of the exposome.** To visualize whether a molecular feature was connected to several exposures, and vice versa, we built period-specific multi-omics exposome networks, based on the 1170 statistically significant exposome-omics associations. The nodes of these networks are the 538 unique molecular features or exposures involved in these associations, and the edges are the 1170 exposure-omics associations.

The pregnancy exposome network, mostly composed of CpGs (70%), was very disconnected having on average 1.3 connexions per node (i.e. degree) and an average shortest path length of 1.9 (Fig. 4 and Supplementary Data 5A). This number represents the average length (number of nodes) of the shortest path between each node and any

other node, 1.9 being a low value. This lack of connectivity can be explained by the wide-spacing along the genome of the CpG sites assessed with the 450 K array and their relatively low correlation. The pregnancy exposome network contained 3 main connected components (referred to as clusters, and labelled "preg#…"), the largest of which contained less than 30% of all nodes. These 3 clusters varied greatly in their size, their number of exposures and the type of omics data comprising them (Table 1).

The childhood exposome network was more densely connected, with an average of 1.9 connexions per node and an average shortest path length of 4.3. The biggest connected component contained 90% of all nodes (Fig. 5). This connectivity highlights the correlated nature of the serum and urine metabolome, which represented the majority of the exposure-omics associations of the network (43 and 26% respectively). Within the biggest connected component, we identified 11 interconnected subcomponents (i.e. clusters, named as "childhood#…") using an unsupervised structural clustering method (Table 1 and Supplementary Data 5B)[28,29].

Next, we aimed to evaluate the biological interpretation of the exposure-omics associations included in the 3 pregnancy and 11 childhood clusters. First, we did a systematic search of overlap with the

**Table 1 | Pregnancy and childhood exposome clusters based on associations with multi-omics profiles measured in childhood (N = 1301)**

| Clusters | Exposures[a] | Omics-associated features | | | | | | | |
|---|---|---|---|---|---|---|---|---|---|
| | | DNA methylation | Gene expression | miRNA expression | Proteins | Serum metab. | Urine metab. | Total | Total annotated genes[b] |
| **Pregnancy period** | | | | | | | | | |
| 1. Tobacco smoke | Cd, cotinine, maternal smoking, Hg, PFUNDA | 39 | 0 | 1 | 0 | 8 | 0 | 48 | 19 |
| 2. Molybdenum | Mo | 72 | 0 | 0 | 0 | 1 | 1 | 74 | 47 |
| 3. Phthalates and parabens | ETPA, MEPA, PRPA, MEHHP, BUPA, MECPP, MEOHP | 1 | 0 | 5 | 0 | 10 | 4 | 20 | 6 |
| **Childhood period** | | | | | | | | | |
| 1. Organochlorine chemicals | PCB 118, HCB, PCB 138, PCB 153, PM$_{2.5}$, PCB 180, house crowding, PCB 170, bread, DDE, diet fat, MEP, MIBP | 43 | 1 | 1 | 3 | 24 | 2 | 74 | 41 |
| 2. Copper | Cu, Pb, land use | 52 | 5 | 17 | 6 | 8 | 1 | 89 | 69 |
| 3. Fish and contaminants | PFOS, Cs, PFNA, Se, PFUNDA, Hg, As, fish, PFHXS, cotinine, dairy, PBDE 153, fastfood, ETS, BTEX in, sweets | 2 | 0 | 3 | 2 | 56 | 8 | 71 | 5 |
| 4. Weather | Hum,, T, UV, K, PFOA, Mg, BPA, Tl, PM$_{10}$ | 30 | 21 | 18 | 5 | 4 | 0 | 78 | 65 |
| 5. Phthalates (DEHP metabolites) | MEOHP, MEHHP, MECPP, MEHP, bakery prod | 8 | 1 | 0 | 0 | 7 | 0 | 16 | 10 |
| 6. Non-persistent chemicals and diet | OXOMINP, KIDMED, DMTP, Co, BUPA, DETP, OHMINP, Cd, fruits, DMDTP, DEP, vegetables, meat, cereals, social participation, DMP | 13 | 0 | 1 | 2 | 1 | 21 | 38 | 15 |
| 7. Indoor air | PM$_{2.5}$ indoor, parental smoking, blue space, MNBP | 13 | 0 | 4 | 0 | 5 | 0 | 22 | 12 |
| 8. Manganese and molybdenum | Mn, Mo | 1 | 6 | 0 | 1 | 2 | 0 | 10 | 8 |
| 9. Benzene | Benzene indoor | 9 | 0 | 0 | 0 | 0 | 0 | 9 | 7 |
| 10. Triclosan | Triclosan | 0 | 0 | 2 | 3 | 0 | 0 | 5 | 5 |
| 11. Distance road | Distance road | 0 | 0 | 0 | 0 | 2 | 0 | 2 | 0 |

$NO_2$ nitrogen dioxide, $PM_{2.5}$ particulate matter with an aerodynamic diameter of <2.5 μm, $PM_{10}$ particulate matter with an aerodynamic diameter of <10 μm, $PM_{2.5abs}$ absorbance of $PM_{2.5}$ filters, *TEX* toluene, ethylbenzene, xylene, *DDE* 4,4'-dichlorodiphenyl dichloroethylene, *DDT* 4,4'dichlorodiphenyltrichloroethane, *HCB* hexachlorobenzene, *PCB* polychlorinated biphenyl—118, 138, 153, 170, 180, *PBDE 47* 2,2',4,4'-tetra-bromodiphenyl ether, *PBDE 153* 2,2',4,4',5,5'-hexa-bromodiphenyl ether, *PFOA* perfluorooctanoate, *PFNA* perfluorononanoate, *PFUNDA* perfluoroundecanoate, *PFHxS* perfluorohexane sulfonate, *PFOS* perfluorooctane sulfonate, *As* arsenic, *Cd* cadmium, *Co* cobalt, *Cs* ceasium, *Cu* copper, *Hg* mercury, *Mn* manganese, *Mo* molybdenum, *Pb* lead, *Tl* thallium, *MEP* monoethyl phthalate, *MiBP* mono-iso-butyl phthalate, *MnBP* mono-n-butyl phthalate, *MBzP* mono benzyl phthalate, *MEHP* mono-2-ethylhexyl phthalate, *MEHHP* mono-2-ethyl-5-hydroxyhexyl phthalate, *MEOHP* mono-2-ethyl-5-oxohexyl phthalate, *MECPP* mono-2-ethyl-5-carboxypentyl phthalate, *OHMINP* mono-4-methyl-7-hydroxyoctyl phthalate, *OXOMINP* mono-4-methyl-7-oxooctyl phthalate, *MEPA* methyl paraben, *ETPA* ethyl paraben, *BPA* bisphenol A, *PRPA* propyl paraben, *BUPA* N-butyl paraben, *OXBE* oxybenzone, *DMP* dimethyl phosphate, *DMTP* dimethyl thiophosphate, *DMDTP* dimethyl dithiophosphate, *DEP* diethyl phosphate, *DETP* diethyl thiophosphate, *DEDTP* diethyl dithiophosphate, *THM* trihalomethane.
[a]Ordered based on their number of omics associations, from the exposures with the most associations to the ones the least connected.
[b]Across CpGs, transcript clusters, miRNA and proteins.

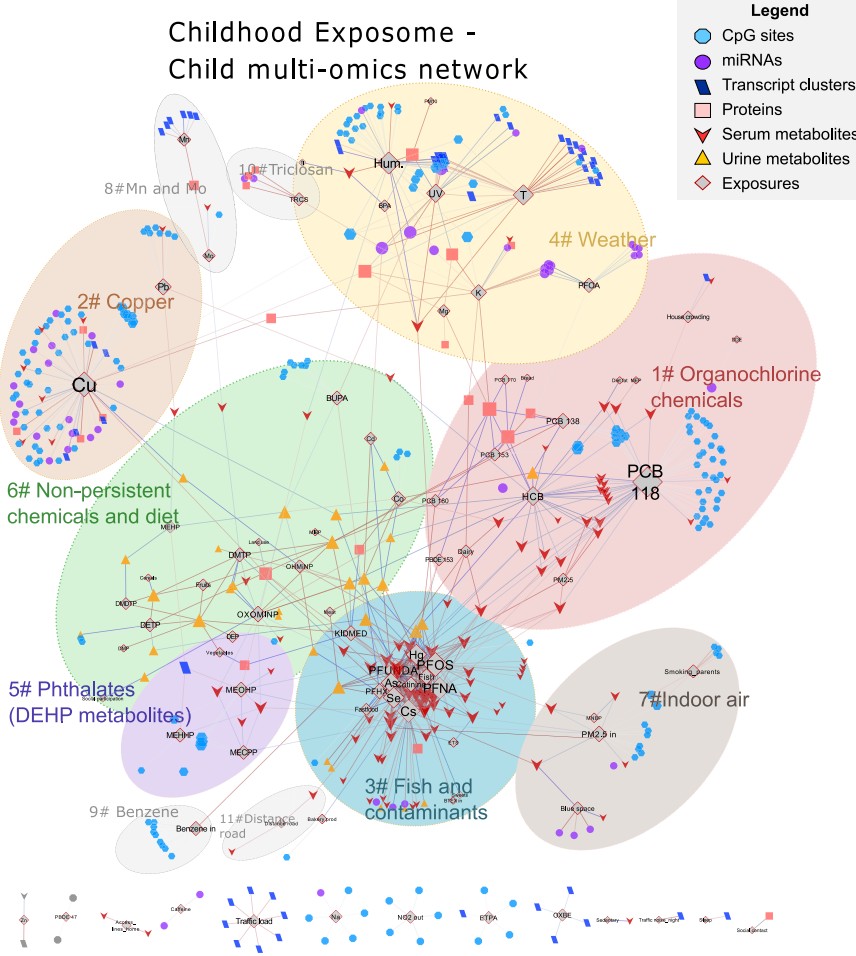

**Fig. 5 | Network map of the multi-omics signatures of the childhood exposome.** Network visualization of the childhood Exposome-omics-Wide Association Study (ExWAS). Nodes of the network are depicted with a different colour/shape depending if they are exposures or features of a particular molecular layer (see legend in the figure). An exposure and a molecular feature were connected with an edge if their association was statistically significant (blue if positively and red if negatively). Only connected components with at least two molecular features were displayed. An exposure and a molecular feature were connected if their association was statistically significant, and only connected components with at least two molecular features were displayed. The childhood exposome network was diverse in terms of omics features represented and the level of interconnection, with the biggest connected component containing 90% of all nodes. Within this network, 11 clusters were identified using an unsupervised structural cluster analysis (see Supplementary Information), and were annotated in the figure. The summary table with the cluster characteristics are in Table 1 and the full table with the node attributes can be found in Supplementary Data 5B.

literature on DNA methylation associations with exposures and traits (EWAS Atlas/Catalogue[16,30], Fig. 6A–C and Supplementary Data 6) and on metabolite associations with dietary patterns and pollutants (ExposomeExplorer[31], Fig. 7). Second, we conducted functional enrichment analyses using several public databases (Fig. 6B–D and Supplementary Data 7). Methodological details can be found in Supplementary Information. Below, we describe the main findings for groups of exposures.

**Maternal smoking shows robust and long-lasting effects in the child methylome and novel signatures for prenatal cadmium and indoor air pollution are detected.** Methylation signatures for maternal smoking at different ages have been well documented[32]. In HELIX, maternal smoking during pregnancy assessed using questionnaires and urinary maternal cotinine levels associated with 24 unique CpGs (cluster preg#1), representing 9 unique loci (2 Mb) annotated to 8 genes, that largely overlap with smoking-sensitive CpGs described in the EWAS Atlas/Catalogue (Fig. 6A–C and Supplementary Data 6). Child exposure to second-hand smoke also overlapped with existing literature, but to a lesser extent than maternal smoking (cluster childhood#7). Period specific smoking effects in HELIX have been investigated elsewhere[25]. Functional enrichment analysis identified the

following pathways: axon development, cognition, cholinergic synapse, insulin signalling, and several types of cancer (Fig. 6B–D and Supplementary Data 7, highlighted in yellow).

Prenatal cadmium (Cd), a heavy metal, was associated with child blood methylation, and mapped with maternal smoking in cluster preg#1. The multi-exposure analyses suggested some overlap between these signals (Supplementary Data 4A). This could be partially explained by the fact that Cd is a component of tobacco[33] and in our dataset mothers who smoked showed almost twice the level of Cd compared to non-smokers. However, we identified 14 additional CpGs that were unique to Cd (Fig. 2A, B and Supplementary Data 8A). When restricting our analysis of maternal Cd to non-smoker mothers ($N = 998$), 51 CpGs (48 loci) were identified (Supplementary Information–Fig. S2C, D and Supplementary Data 8B). These did not overlap with known smoking effects, nor with CpGs associated with urinary Cd in adult blood or with placental Cd in placental tissue[34,35].

We further found several associations for air quality during childhood, which did not overlap between outdoor and indoor pollutants. Among the most interesting, home indoor air pollution exposure to benzene was associated with 9 CpGs (cluster childhood#9), one of them related to PM$_{2.5}$ levels in previous studies (Fig. 6C and Supplementary Data 6). Moreover, home indoor levels of PM$_{2.5}$ absorbance, a

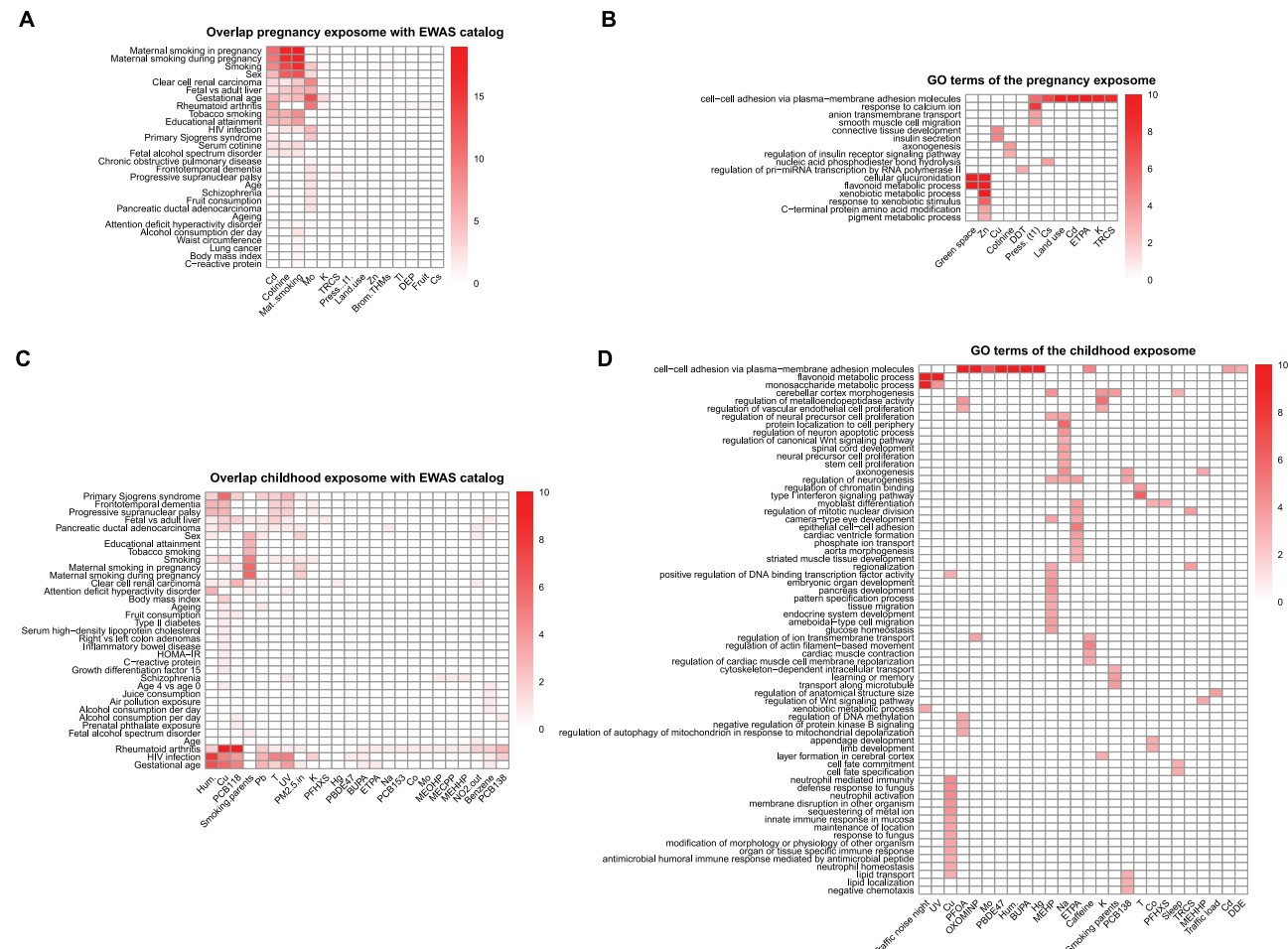

**Fig. 6 | Biological interpretation of the exposome-omics associations through literature overlap and functional enrichment. A** Overlap of CpGs associated with the pregnancy exposome (columns) with CpGs associated with traits/exposures in the EWAS catalogue (rows). **B** Functional enrichment analyses of the pregnancy exposome (columns) for Gene Ontology (GO) terms (rows). **C** Overlap of CpGs associated with the childhood exposome (columns) with CpGs associated with traits/exposures in the EWAS catalogue (rows). **D** Functional enrichment analyses of the childhood exposome (columns) for GO terms (rows). Exposure variables, traits/exposures of the EWAS catalogue, and GO terms are ordered according to a hierarchical clustering. For the overlap with the EWAS catalogue, colour indicates the number of overlapping CpGs. For the functional enrichment analyses, colour indicates the −log10 adjusted *p* value of the enrichment. To facilitate visualization, we eliminated related GO terms and −log10 adjusted *p* values >10 are coded as 10.

marker of black/elemental carbon originating from combustion, were associated with methylation of 9 CpGs, including two in common with tobacco exposure (Fig. 6C and Supplementary Data 6), and with decreased levels of serum branched amino acids (BCAA), C4 acylcarnitine and two sphingolipids (cluster childhood#7). Some of these associations were attenuated after adjusting for parental smoking (Supplementary Data 4B).

**The serum and urinary metabolome reveal principal dietary routes of exposure to chemical pollutants.** Cluster childhood#3 contained fish intake (information collected through questionnaire), toxic metals (mercury (Hg) and arsenic (As)), the per- and polyfluoroalkyl substances (PFAS), and non-toxic essential elements (selenium (Se) and caesium (Cs)), together with serum lipids containing polyunsaturated fatty acids (PUFA) and urinary trimethylamine *N*-oxide (TMAO), dimethylamine and homarine (Fig. 7A). Using systematic metabolite-diet associations found in previous population studies archived in ExposomeExplorer[36], we confirmed the dietary origin of these exposure-metabolite associations, in this case to fish intake and animal products (Fig. 7C). In addition, multi-exposure models confirmed that most of these associations in particular with Hg, As and PFAS were attenuated after adjusting for diet and other co-exposures. This was

not true for TMAO and As which remained one of the strongest association even after adjusting for PCB 180, Hg, Fish and PFUNDA (Supplementary Data 4B).

Similarly, cluster childhood#6 contained 21 out of the 44 urinary metabolites measured, including hippurate, proline betaine and *N*-methylnicotinic acid which are known biomarkers of fruit and vegetable intake[26,37] (Fig. 7B, C). The cluster also included organophosphate (OP) pesticides measured in urine which suggested a potential route of exposure through dietary intake of fruits and vegetables.

Also in cluster childhood#6, we found the DiNP metabolites, phthalate family members primarily used to produce flexible plastics such as food packaging. In contrast, DEHP metabolites (MEOHP, MEHHP, MECPP, MEHP), also phthalates found in plastics, mapped in cluster childhood#5 and were associated with 13 CpGs, with no clear overlap with reported traits/exposures (Fig. 6C and Supplementary Data 6). MEOHP and MECPP were also negatively associated with a number of serum sphingomyelins (SM C16:0, SM C18:0, SM C18:1, SM C20:2, SM (OH) C14:1 and SM (OH) C16:1). Pregnancy exposure to DEHP metabolites and parabens, synthetic compounds present in personal care products, also showed negative associations with sphingomyelins (SM (OH) C16:1) and valine in children.

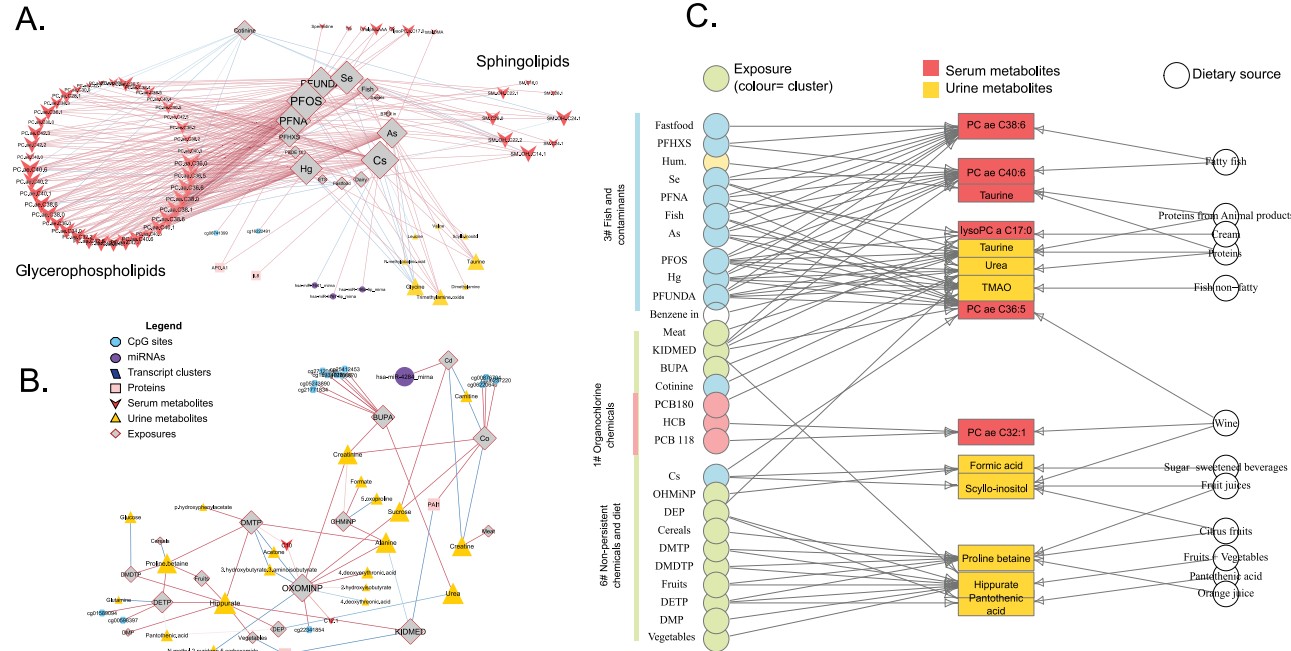

**Fig. 7 | Metabolite signatures of the childhood exposome and dietary sources.**
**A** Cluster childhood#3 includes the significant associations between fish and several contaminants (As, Hg, PFOS) and serum metabolites (mainly glycerophospholipids). **B** Cluster childhood#6 includes the significant associations between diet (vegetables, fruit, cereals) and organophosphate (OP) pesticides with urinary metabolites. For **A**, **B**, nodes of the network are depicted with a different colour/shape depending if they are exposures or features of a particular molecular layer (see legend in the figure). An exposure and a molecular feature were connected with an edge if their association was statistically significant (blue if positively and red if negatively). **C** Tripartite plots based on the presence of associations between metabolites-exposure in HELIX samples (on the left) and metabolites-dietary intake based on the ExposomeExplorer database (http://exposome-explorer.iarc.fr/)[36] (on the right). Serum and urinary metabolites are shown in red and yellow, respectively, and exposures in blue (fish and contaminants), red (organochlorine chemicals) or green (non-persistent chemicals and diet) according to the cluster they belong to.

**Essential trace elements are key components of the exposome.**
Essential trace elements are required by living organisms to ensure normal development and maintenance of biological functions, but can also be toxic when present in excess. We measured 9 essential elements in whole blood (Co, Cu, Mn, Mo, Na, K, Mg, Zn, Se), and found a remarkable number of exposure-omics associations, mostly with maternal molybdenum (Mo), and child copper (Cu) (Supplementary Data 2).

Maternal Mo was related to the methylation levels of 72 CpGs, representing 63 loci (cluster preg#2). No relevant gene-sets were identified for genes annotated to these 72 CpGs, but 13 of them have previously been related to gestational age (Fig. 6A and Supplementary Data 6). Mo acts as a co-factor of 4 human enzymes which are involved in various key reactions, including the regulation of sulfur-containing amino acids such as methionine[38]. In our dataset, maternal Mo was associated with higher methionine levels in childhood (Supplementary Data 2).

Child Cu was associated with 89 molecular features, distributed across different omics layers (cluster childhood#2). One of the associations with the lowest $p$ value was with increased levels of the C-reactive protein (CRP), a marker of inflammation. Moreover, Cu-associated CpGs have previously been linked to obesity, type 2 diabetes and rheumatoid arthritis, a chronic inflammatory disorder, among others (Fig. 6C and Supplementary Data 6). Enriched pathways for Cu included: immune response, lipid storage and sequestering of metal ions (Fig. 6D and Supplementary Data 7, highlighted in green). Adjusting for co-exposures (e.g. Pb) did not change substantially these associations (Supplementary Data 4B).

Furthermore, during childhood, other essential trace elements were associated with multiple molecular features with no overlap among them (Supplementary Data 2), as expected due to their intrinsic essential roles. For instance, zinc (Zn) was related to higher transcription of *CA1* (*Carbonic anhydrase 1*), whose expression is known to be influenced by Zn²⁺ availability and which uses $Zn^{2+}$ as a cofactor for its enzymatic activity[39].

**Weather conditions are associated with signatures in all omics layers.**
Weather conditions or meteorological factors (temperature, humidity, cloud coverage and atmospheric pressure), in particular when extreme, are strong determinants of health and mortality[40]. However, there are no studies systematically assessing their influence on molecular phenotypes. We estimated weather conditions through geographical information coupled with data from meteorological stations (Supplementary Information). In childhood, weather conditions over the month before the omics measurement, were associated with all molecular layers, except for the urinary metabolome (cluster childhood#4). Serum metabolites associated with meteorological variables included taurine, asymmetric dimethylarginine (ADMA), acylcarnitine C5, and serotonin, which have been previously reported as biomarkers of sleep deprivation, circadian rhythm and in the aetiology of depression[41–43] (Supplementary Data 2). They were also associated with three proteins: adiponectin, MCP1 and HGF. Adiponectin, an essential regulator of thermogenesis[44,45], increased with humidity (higher in winter in Europe) and decreased with ultraviolet radiation (higher in summer) (Supplementary Data 2). This is in line with previous studies showing that exposure to cold temperatures for 2 h increases adiponectin plasma levels[46]. The magnitude of some of these associations (carnitine C5, adiponectin, serotonin) were attenuated by more than 50% after adjusting for exposure to bisphenol A (BPA), which was previously found to reduce adiponectin release[47] (Supplementary Data 4B). Finally, the CpGs associated with weather conditions overlapped with CpGs reported for infections, among others (Fig. 6C and Supplementary Data 6); and genes related to temperature were enriched for cellular response to type I interferon

(Fig. 6D and Supplementary Data 7, highlighted in blue). Infectious diseases follow seasonal patterns and are more prevalent under particular meteorological conditions, as recently shown with in the different COVID-19 pandemic waves[48,49].

**Persistent organic pollutants (POPs) and multi-omics alterations in children.** We found that POPs in children, especially dioxin-like PCB 118 (69 associations), HCB (28) and PCB 138 (14), were associated with DNA methylation, serum metabolites and plasma proteins (IL1B and leptin) grouped in cluster childhood#1 (Fig. 5). CpGs in this cluster have previously been reported to be related to the inflammatory disease rheumatoid arthritis (Supplementary Data 6), and IL1B and leptin are produced by the fat tissue as commented above. We also observed an unique positive association of PCB 180 and urinary TMAO, without any other associations with other fish-related metabolites described above.

**Replication of exposure-omics associations across molecular layers and biological matrices.** We investigated whether childhood exposome associations with DNA methylation, gene and miRNA expression, all assessed in blood cells, pointed to the same genes. For each CpG, we identified *cis* expression quantitative trait methylations (eQTMs), meaning correlations between gene expression and DNA methylation (Supplementary Information). Out of the 187 CpGs associated with the childhood exposome, 9 had eQTMs in a total of 11 genes (Supplementary Data 9A). However, none of these eQTMs was nominally associated with the same exposures as the CpG site. We also searched for targeted genes of the 49 miRNAs associated with the childhood exposome using the miRwalk v3 tool[50] (Supplementary Data 9B). Seventeen out of the 1267 targeted genes were associated with the same exposure as the original miRNA and in the expected direction (higher miRNA levels – lower gene expression). They encompassed 7 unique exposures (Cd, Cu, K, PFOA, blue spaces and meteorological factors) and 9 unique miRNAs (Supplementary Data 9C).

We also compared the overlap of childhood exposure associations for 12 metabolites (amino acids, glucose, carnitine and creatinine) that were measured in both urine and serum, and whose correlation can be found in Supplementary Data 10A. At nominal significance, 27.3% of the urine associations replicated in serum; and 7% of the serum associations replicated in urine (Supplementary Data 10B, C). Not surprisingly, replicated associations involved metabolites with the highest correlation between matrices (carnitine, glycine and creatinine) (Supplementary Data 10A).

## Discussion

This is the first exposome study to systematically associate a wide range of environmental exposures during vulnerable early life periods with multi-omics signatures in childhood. We observed 1170 unique associations between exposures and molecular features, 249 relating to pregnancy and 921 to childhood exposures. By partitioning these associations into network clusters for visualization and by conducting systematic biological interpretation, this study reveals potential biological responses and sources of exposure. Our findings confirm persistent methylation changes associated with maternal tobacco smoking in pregnancy[51] and principal sources of exposure to chemical pollutants through diet, based on food-related biomarkers. Furthermore, we identify novel associations notably with essential trace elements, weather conditions, indoor air quality, persistent pollutants, phthalates and parabens. Our comprehensive resource of all associations (https://helixomics.isglobal.org/) is the first of its kind and will serve to guide future investigation on the biological imprints of the early life exposome.

Our web catalogue has several applications: creating biomarkers of exposure, identifying sources of exposures and understanding biological mechanisms. Data generated in this study provide a resource for the development of epigenetic biomarkers of past exposures[52]. For instance, it was generally believed that the essential element molybdenum (Mo) is safe for human health[53]; however, there is growing evidence that excess of Mo is associated with developmental effects and with adverse health outcomes[20,54–58]. In this study, maternal levels of Mo were associated with methylation changes in a remarkable number of CpGs, which were persistent at least until childhood (when we detected them). The methylation in these CpGs could be used to predict prenatal exposure levels.

Also, our study demonstrates the ability of metabolomics to accurately reflect dietary sources and potential gut microbial effect of exposures. The strongest, most significant associations among all exposome-omics tested were found for As and Hg with trimethylamine-*N*-oxide (TMAO) and glycerophospholipids. Most of these associations, except the TMAO-As association, were attenuated after adjusting for fish intake and other fish-related compounds. Indeed, TMAO was previously demonstrated to discriminate high against low fish intake, whereas homarine (a metabolite found in shellfish muscle) for high/non shellfish intake in populations with high seafood intake such as in Spain and Japan[59,60]. TMAO–As association that remained the strongest association after adjusting for fish related exposures also suggests the independent role of the gut microbiome. This finding corroborates our previous study in pregnant women from the Spanish INMA cohort[59]. Other evidence indicate that gut microbiome may alter arsenic metabolism and neurodevelopmental susceptibility to this exposure[61,62]. Importantly, we illustrate in this study that many anthropogenic chemicals are delivered to the body through diet (in this case fruit and fish intake), which biological effect may be altered by the gut microbiome, adding to the complexity of metabolomic profiles in human biospecimens and creating an extensive network of nutrient–pollutant interactions that remains vastly unknown and poorly defined by conventional assessment methods[63].

Among the novel molecular signatures identified, six groups of exposures highlighted plausible biological mechanisms to disease. First, Cu is an essential trace element required for numerous cellular processes, including mitochondrial respiration, antioxidant defence, neurotransmitter synthesis, and iron metabolism, among others[64]. In previous HELIX studies, Cu has been related to several health outcomes such as poorer lung function[65], higher BMI[66,67] and blood pressure[68], and increased ADHD symptomatology[69], and here we show potential perturbed pathways that may mediate these associations: immune response, lipid storage and sequestering of metal ions. Second, pathways identified for tobacco smoke (axon development, cognition, cholinergic synapse, insulin signalling, and several types of cancer) were similarly in line with the effects of maternal smoking on health outcomes detected in HELIX children (higher blood pressure[68] and BMI[66], and increased behavioural problems[69]). We acknowledge that, as DNA methylation was measured in blood, the identification of pathways relevant for other tissues (i.e. brain and axon development) has to be analysed with caution. It could be that DNA methylation marks are maintained across tissues if exposure happens early in development, or that the same genes are involved in different pathways in different tissues. Third, indoor air quality during childhood was associated with metabolic markers (BCAA and acylcarnitines). The HELIX study was the first to find an association between indoor air pollution and child obesity[66]. Dysregulated metabolism of BCAAs and acylcarnitines has been associated with obesity and insulin resistance in numerous studies[70] and was detected in young obese participants exposed to near-roadway air pollution[71]. Altered BCAA and acylcarnitine metabolism may be an important biomarker to study further in relation to air pollution and cardio-metabolic disease risk in later life. Fourth, POPs have consistently been associated with adverse heath outcomes[72,73]. Besides associations likely linked to fat distribution in children, we also observed a positive association of PCB 180 and

TMAO, a product of gut microbiota and liver hepatic flavin containing monooxygenase (FMO3) enzyme activity. This association was previously reported in animals and humans and appeared independent of potential common dietary sources of PCBs and TMAO, and of BMI[74]. Currently, TMAO is proposed as a causative agent of cardio-vascular disease[75] but further investigations on the mechanistic link between PCBs, FMO3 activity/expression and cardio-vascular outcomes are needed. Fifth, we found associations with high molecular weight phthalates and parabens, which are synthetic compounds rapidly metabolized in the body and suspected of being endocrine disruptors[76] and affecting health in a sex-specific manner[24]. Exposure to phthalates occurs mostly through diet, dust ingestion, and to a less extent through inhalation[77]. Metabolic signatures of phthalates and parabens were not clearly related to dietary patterns but to an endogenous metabolic pathway, the sphingomyelins, which are important structural lipid components of cell membranes involved in signalling and implicated in many disorders[78,79]. Intermediates of sphingosine biosynthesis and valine have been reported to be upregulated in pregnant women exposed to phthalates[80] and parabens[81]. Sixth, our results also provide insights into potential mechanisms of action for weather conditions: they appear to have direct effects (e.g. regulating thermogenesis) and indirect effects (e.g. determining other exposures such as virus survival), or they can also represent proxies of other variables (e.g. hours of daylight or dietary changes due to seasonal variation). The investigation of meteorological conditions in larger longitudinal omics datasets covering seasonal patterns will be needed to elucidate the final causal mechanisms.

Our study indicates that the choice of molecular layer and biological matrix is key in the design of exposome studies. Most of the associations we found for the pregnancy exposome involved the methylome (70% of the associations observed). This is in line with previous publications that suggest that the epigenome acts as the main source of cellular 'memory' and plasticity[82,83]. Although, it may partially reflect the nature of our study design and omics coverage (i.e. number of markers analysed in each omics layer and their intra-omics correlations). In contrast, recent exposures during childhood were associated with features across all omics layers. Evidence to date suggests that the metabolome in particular is strongly influenced by the immediate environment, and may thus be more sensitive for detecting associations in cross-sectional settings[17]. Nevertheless, many cross-sectional associations with the methylome were found and, although fewer, long-term associations with other omics were also found. Moreover, the low correspondence between the methylome and miRNAome with the transcriptome highlights the high complexity of transcriptional regulation and suggests that each molecular layer might capture a window of the effects of the exposome. Our findings also indicate the importance of the biological matrix. Although we could not make a comprehensive comparison of the urinary and sera metabolomes because of the use of different platforms to assess them, among comparable metabolites, only a few showed consistent associations with the exposome in both biological matrices. Thus, both biological matrices and others should ideally be explored in exposome studies, providing complementary information. Finally, we observed little overlap in associations for the pregnancy and childhood exposome, likely due to the low inter-period correlation of exposures, the differences in the exposure route or dose between periods, and the dynamics of the molecular response (i.e. our study is able to capture long-term responses of the pregnancy exposome but only short-term responses of the childhood exposome). This highlights the importance of the windows of exposure and the choice of life course framework for exposome studies.

Our study has multiple strengths. First, the comprehensive assessment of environmental exposures in two critical developmental time periods, including highly sensitive biomarkers for many chemical exposures and wide-ranging geospatial modelling of the outdoor and built environment. Second, the extensive multi-omics assessment of molecular phenotypes. Third, the wide geographic coverage and relatively large sample size for which we were able to measure many exposures and omics features. Finally, we conducted several sensitivity analyses, that confirmed that findings were robust to ancestry and zBMI, with the exception of some lipophilic exposure compounds and particular molecular features.

Our study also has some limitations. First, omics platforms have a coverage bias and biological interpretability issues. For instance, the LC-MS/MS (Biocrates) method has a low coverage and does not give specific fatty acid side-chain composition for lipids, but it is widely used in large cohort studies and provides reproducible measurements with unambiguous annotation, easily comparable to other studies[84–88]. We note that there are additional molecular layers and omics technologies of interest for future exposome studies, which were not included in our study, such as the gut metagenome, sensitive high-resolution mass spectrometry or single cell methods[89–91]. Moreover, the effect of genetic variation, alone or in combination with the exposome, was not considered in this study. Second, different exposures are measured with different types and levels of measurement error. For example, urine levels of non-persistent chemicals have a high intra-individual variability and are expected to suffer particularly from classical-type measurement error resulting in an attenuation bias[92]. Repeated sampling strategies and longitudinal designs, might help to disentangle the persistent metabolic effects of endocrine disruptors suggested in our study. Exposures measured by models and questionnaires are expected to suffer from other types of measurement errors with less predictable effects[93]. Moreover, the correlated nature of the exposome makes identification of driving exposures difficult. Here we tried to separate the effects with mutually adjusted or stratified models. For example by running stratified models in non-smoker mothers we identified Cd-specific effects. Besides tobacco smoke, Cd might have other origins such as rice, potatoes and wheat, when frequently consumed in large quantities[94]. Mixture or multi-pollutant approaches aim to tackle this more systematically, however these are not yet suitable for high-dimensional omics datasets such as ours[95,96]. Third, our comparison with previous literature and functional enrichment analyses are limited by existing bias in public databases. Fourth, although the majority of epidemiological studies utilize biological samples that are most readily accessible for the measurement of omics profiles, these may not be the ideal target tissue for the relevant health outcomes. Fifth, some associations presented high heterogeneity across cohorts (e.g. humidity and serotonin). This can be explained by the different exposure levels, the different correlation with confounders, or the relatively small sample size within each cohort. Finally, although our models were adjusted for confounders, residual confounding might still be present and causal links would need to be proven through interventions, Mendelian randomization analyses, cross-contextual studies, or in vivo/in vitro models.

To conclude, this first comprehensive study of the multi-omics signatures of the early life exposome demonstrates that molecular phenotypes can reveal biological responses to or sources of environmental exposures at an early time point in life. Besides the main findings described here, the entire result catalogue is publicly available (https://helixomics.isglobal.org/), enabling exploration of the complete list of exposome-omics relationships. With the rich exposome and molecular information available, we provide a valuable resource to the scientific community for the development and validation of exposure and response biomarkers, to identify dietary sources of exposures, to improve our understanding of disease aetiology, and finally to promote public health policies.

## Methods
Local ethical committees approved the studies that were conducted according to the guidelines laid down in the Declaration of Helsinki.

The ethical committees for each cohort were the following: BIB: Bradford Teaching Hospitals NHS Foundation Trust, EDEN: Agence nationale de sécurité du médicament et des produits de santé, INMA: Comité Ético de Inverticación Clínica Parc de Salut MAR, KANC: LIE-TUVOS BIOETIKOS KOMITETAS, MoBa: Regional komité for medisinsk og helsefaglig forskningsetikk, Rhea: Ethical committee of the general university hospital of Heraklion, Crete. Informed consent was obtained from a parent and/or legal guardian of all participants in the study. Participants did not receive any compensation.

## Population

Mother–child pairs (N = 1301) from 6 established and ongoing longitudinal population-based birth cohort studies in Europe were included in the HELIX subcohort study: the Born in Bradford (BiB) study in the UK[97], the Étude des Déterminants pré et postnatals du développement et de la santé de l'Enfant (EDEN) study in France[98], the INfancia y Medio Ambiente (INMA) cohort in Spain[99], the Kaunus cohort (KANC) in Lithuania[100], the Norwegian Mother, Father and Child Cohort Study (MoBa)[101], and the RHEA Mother Child Cohort study in Crete, Greece[102] (Supplementary Information and Supplementary Data 1A). A follow-up examination of the children between ages 6 and 11 years was carried out with fully standardized protocols across the six cohorts, in order to assess child health outcomes, to fully characterize the pregnancy and childhood exposome, and to measure several molecular phenotypes[21]. During the clinical examination, urine (pooled spot urine samples from before bedtime and first morning void) and blood samples were collected from the children. Urine and blood samples previously collected from mothers during pregnancy were also available for biomarkers of chemical exposure assessment.

## Exposome measures in pregnancy and childhood

Two main windows of exposure were considered: a prenatal window including the pregnancy period or measures of long-term maternal exposures (e.g. persistent pollutants), and a cross-sectional window including the exposome data of children at the same time as of omics sampling (childhood). A total of 91 pregnancy and 116 childhood exposures were investigated in the study, including the outdoor exposome (air pollution, built environment, noise, green and blue space, and meteorological data), the chemical exposome (cotinine, metals, POPs, PFAS, phthalates, phenols, and organophosphates), and social and lifestyle factors (exposure to tobacco smoking, diet and physical activity). Details on the exposure assessment methods and exposure factors can be seen in Supplementary Information. Exposures were either continuous variables or categorical variables with two or more levels. Continuous exposure variables were transformed to achieve linearity or categorized, when needed. Missing data were imputed using a chained equations method[103] implemented in the mice v3.4.0 R package[104]. One imputed dataset was used in this study. Further details on exposure levels can be found elsewhere[22–24]. Correlations between exposures were estimated as follows: for continuous vs continuous variables—Pearson's correlation; for continuous vs categorical variables—$R^2$ of a lineal model; for categorical vs categorical variables—Cramér's V test. More information can be found in Supplementary Information and Supplementary Data 1C–E.

## Child molecular phenotypes

We used both targeted and untargeted methods to assess child molecular phenotypes. Blood DNA methylation was assessed with the Illumina 450 K array; blood gene expression, with the Affymetrix HTA v2.0 array; blood miRNA expression, with the Agilent SurePrint Human miRNA rel 21 array; plasma proteins, with 3 Luminex multiplex assays; serum metabolites, with the targeted LC-MS/MS metabolomic assay Biocrates AbsoluteIDQ p180 kit; and urinary metabolites, with $^1$H nuclear magnetic resonance (NMR) spectroscopy. An extended version of the omics protocols and lists of biomarkers assessed in the targeted assays is available in Supplementary Information and Supplementary Data 1E–H.

## Statistical analysis (ExWAS)

We fitted linear regressions between each exposure variable and each molecular feature adjusting for covariates, using the limma v3.46.0 R package[105] implemented in omicRexposome v1.12.1[106]. Main covariates for all omics were: cohort, child's sex, child's age, child sex and age z-score BMI calculated according to WHO reference curves[107,108], child's ethnicity defined in three categories (European ancestry; Pakistani or Asian; and other), and self-reported maternal education categorized in low, medium and high. In addition, plasma protein, serum metabolite models were adjusted for time to last meal and hour of blood collection and urinary metabolite models for sample type (bedtime, morning or pool), and technical batch. Blood methylation and transcriptomics data were corrected by surrogate variables (SVs), which captured both batch effects and blood cell type composition.

In all omics, except for methylation, the effect size is reported as a log2 fold change (log2FC) of the molecular phenotype levels between categories of discrete exposure variables or for interquartile range (IQR) of continuous exposure variables. For DNA methylation, the effect size is reported as a difference in methylation levels between categories of discrete exposure variables or for IQR of continuous exposure variables.

Multiple testing correction was applied for each exposure and within each omics layer. For methylation, gene expression and miRNAs we used the False Discovery Rate (FDR)–Benjamini–Hochberg (BH) method[109]. For other omics, proteins, urine and serum metabolites, we calculated the effective number of tests (ENT) which is based on the correlation structure of the data[110], and the nominal p value (0.05) divided by that number. We also calculated a more stringent threshold correcting for all tests performed (across all molecular features from all omics platforms and the full exposome, including both periods), resulting in a p value cut off of 1E−09. More details can be found in Supplementary Information and Supplementary Data 1I.

## Sensitivity analyses

A set of sensitivity analyses were conducted. First, analyses were restricted to children of European ancestry (90%). Second, models were run again without adjustment for child zBMI. The difference in the effect size among main models and alternative models was calculated as (effect size main model − effect size alternative model)/effect size main model × 100. Third, top hit associations were run by cohort and combined through fixed- and random-effects inverse variance weighted meta-analyses using the meta v4.16-1 R package[111], and forest-plots were visually inspected. $I^2$ was used to evaluate heterogeneity in the results across cohorts. Fourth, we performed multi-exposure linear models by period for those omics features associated with more than one exposure, when these exposures had a correlation <0.8 and belonged to different exposure families (except individual exposures that belonged to diet, metals or parabens that we considered as separate groups).

## Exposure-omics network analyses

We conducted network analyses using the list of ExWAS associations passing multiple testing correction (1170). We built a network for each exposure period (period-specific), and each network contained all the molecular layers (multi-omics). Molecular features and exposures were considered as the nodes of the network and the edges represented omics-exposure associations (based on the ExWAS results). Networks visualization was carried out using Cytoscape 3.6.1 (http://cytoscape.org) and were automatically arranged using the Cytoscape force-directed layout which aims to highlight the underlying topology of the graph[112]. The association effect size was set as the numeric edge column to use as a weight for the length of the edges. In order to find

densely connected regions in the network, clustering of the childhood network was done based on Community Clustering (GLay) using clusterMaker2 v2.0[28,29].

## Comparison with literature

Molecular features of significant associations were checked against previous literature findings based on existing databases reporting associated exposures or traits: the EWAS Catalogue (http://ewascatalog.org/)[30], the EWAS Atlas (http://bigd.big.ac.cn/ewas/index)[16] and the Exposome Explorer database (http://exposome-explorer.iarc.fr/)[31,36]. More details can be found in Supplementary Information and Supplementary Data 1J.

## Functional enrichment analyses

Functional enrichment analyses were restricted to molecular layers with features which could be easily annotated at the gene level: DNA methylation, gene and miRNA expression, and proteins. For exposures with at least one significant association, we retrieved all molecular features associated at $p$ value <1E-03. Then, we annotated these molecular features to genes as described in Supplementary Information, and obtained a unique list of "dysregulated" genes by combining genes detected in any of the molecular layers. ClusterProfiler v3.8.0 R package[113] was used to check whether this list of genes was enriched for gene-sets (Gene Ontology (GO) Biological Processes terms, KEGG, Molecular Signatures Database–C2 curated gene sets), diseases (DisGeNET), and transcription factor and miRNA binding motifs (Molecular Signatures Database–C3 motifs and transcription factors motifs). Multiple-testing was corrected with the FDR–BN method within each exposure and only gene sets with >3 genes are reported.

## Expression quantitative trait methylation (eQTMs) and miRNA gene target prediction

To identify experimentally validated target genes for miRNAs we used miRwalk v3[50]. Expression quantitative trait methylations in *cis* (*cis*-eQTMs) were identified using HELIX data. First, we paired each transcript cluster (TC) to all CpGs closer than 500 kb from its transcription start site (TSS), and then, for each CpG-TC pair we fitted a linear regression model between gene expression and methylation levels adjusted for age, sex and cohort. More details on the analyses and the multiple-testing correction can be found in Supplementary Information and elsewhere[114].

## Reporting summary

Further information on research design is available in the Nature Research Reporting Summary linked to this article.

## Data availability

The summarized results (exposure, omics biomarker, effect, standard error, $p$ value) generated during this study are available at https://helixomics.isglobal.org/. The raw data supporting the current study are available from the corresponding author on request subject to ethical and legislative review. The "HELIX Data External Data Request Procedures" are available with the data inventory in this website: http://www.projecthelix.eu/data-inventory. The document describes who can apply to the data and how, the timings for approval and the conditions to data access and publication.

## Code availability

The code to test the relationship between the pregnancy and childhood exposomes and molecular features is available through the omicRexposome v1.12.1 R package[106].

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

## Acknowledgements

We would like to thank all the families for their generous contribution. The study has received funding from the European Union's Horizon 2020 research and innovation programme under grant agreement No 874583 (ATHLETE project). Data were collected as part of the European Community's Seventh Framework Programme (FP7/2007-206) under grant agreement no 308333 (HELIX project). BiB received core infrastructure funding from the Wellcome Trust (WT101597MA) and a joint grant from the UK Medical Research Council (MRC) and Economic and Social Science Research Council (ESRC) (MR/N024397/1). INMA data collections were supported by grants from the Instituto de Salud Carlos III, CIBERESP, and the Generalitat de Catalunya-CIRIT. KANC was funded by the grant of the Lithuanian Agency for Science Innovation and Technology (6-04-2014_31V-66). The Norwegian Mother, Father and Child Cohort Study is supported by the Norwegian Ministry of Health and Care Services and the Ministry of Education and Research. The Rhea project was financially supported by European projects (EU FP6-2003-Food-3-NewGeneris, EU FP6. STREP Hiwate, EU FP7 ENV.2007.1.2.2.2. Project No 211250 Escape, EU FP7-2008-ENV-1.2.1.4 Envirogenomarkers, EU FP7-HEALTH-2009- single stage CHICOS, EU FP7 ENV.2008.1.2.1.6. Proposal No 226285 ENRIECO, EU- FP7- HEALTH-2012 Proposal No 308333 HELIX), and the Greek Ministry of Health (Program of Prevention of obesity and neurodevelopmental disorders in preschool children, in Heraklion district, Crete, Greece: 2011-2014; "Rhea Plus": Primary Prevention Program of Environmental Risk Factors for Reproductive Health, and Child Health: 2012-15). ISGlobal acknowledges support from the Spanish Ministry of Science and Innovation through the "Centro de Excelencia Severo Ochoa 2019-2023" Program (CEX2018-000806-S), and support from the Generalitat de Catalunya through the CERCA Program. L.M. is funded by a Juan de la Cierva-Incorporación fellowship (IJC2018-035394-I) awarded by the Spanish Ministerio de Economía, Industria y Competitividad. M.V.-U. and C.R.-A. were supported by a FI fellowship from the Catalan Government (FI-DGR 2015 and #016FI_B 00272). M. Casas received funding from Instituto Carlos III (Ministry of Economy and Competitiveness) (CD12/00563 and MS16/00128).

## Author contributions

O.R., L.M., M.B., J.W., R.S., J.S., C.T., M. Coen, J.R.G., H.K. and M. Vrijheid designed the omics study in HELIX. The following authors participated in omics data acquisition and quality control: A.C., I.Q., M.B., C.R.-A. (DNA methylation), X.E., M.B., M.V.-U. (transcriptomics), E.S., E.B., M.B., J.R.G. (proteomics), C.E.L., A.P.S., L.M., H.K., M. Coen (metabolomics). O.R., J.W., D.M., R.S., B.H., S.A., J.S., M. Casas, K.B.G., E.P., R.G., L.C., C.T, M. Vafeiadi and A.K.S. are the PIs of the cohorts or participated in sample and exposure data acquisition. M. Casas, C.T., A.K.S., M.N. and I.T. measured the pregnancy and postnatal exposomes. C.H.-F., L.M., M.B., D.T., S.C., J.U., D.P.-S. and J.R.G. performed statistical analyses and J.R.G. functional enrichment analyses. The HELIX project was coordinated by M. Vrijheid. L.M., M.B. and M. Vrijheid wrote the original draft of the paper and C.H.-F., A.P.S., O.R., J.W., D.M., S.C., K.B.G., E.P., C.T., A.K.S., M.N., J.U., M. Coen and H.K. contributed to reviewing and editing the manuscript. All authors read and approved the manuscript.

## Competing interests

The authors declare no competing interests.

## Additional information

Léa Maitre [1,2,3,21], Mariona Bustamante [1,2,3,4,21], Carles Hernández-Ferrer [1,2,3], Denise Thiel[5], Chung-Ho E. Lau [6,7], Alexandros P. Siskos [8], Marta Vives-Usano[1,2,3,4], Carlos Ruiz-Arenas [1,2,3], Dolors Pelegrí-Sisó [1,2,3], Oliver Robinson [6], Dan Mason [9], John Wright[9], Solène Cadiou[10], Rémy Slama[10], Barbara Heude [11], Maribel Casas [1,2,3], Jordi Sunyer[1,2,3,12], Eleni Z. Papadopoulou[13], Kristine B. Gutzkow[13], Sandra Andrusaityte [14], Regina Grazuleviciene [14], Marina Vafeiadi [15],

Leda Chatzi[16], Amrit K. Sakhi[13], Cathrine Thomsen ®[13], Ibon Tamayo[17], Mark Nieuwenhuijsen[1,2,3], Jose Urquiza ®[1,2,3], Eva Borràs ®[2,4], Eduard Sabidó[2,4], Inés Quintela[18], Ángel Carracedo[18,19], Xavier Estivill[4], Muireann Coen[7,20], Juan R. González ®[1,2,3,21], Hector C. Keun ®[7,8,21] & Martine Vrijheid ®[1,2,3,21] ✉

[1]Institute for Global Health (ISGlobal), Barcelona, Spain. [2]Universitat Pompeu Fabra (UPF), Barcelona, Spain. [3]Consorcio de Investigación Biomédica en Red de Epidemiología y Salud Pública (CIBERESP), Madrid, Spain. [4]Center for Genomic Regulation (CRG), Barcelona Institute of Science and Technology, Barcelona, Spain. [5]Department of Mathematics, Imperial College London, South Kensington, London, UK. [6]MRC Centre for Environment and Health, School of Public Health, Imperial College London, London, UK. [7]Division of Systems Medicine, Department of Metabolism, Digestion and Reproduction, Imperial College London, London, UK. [8]Cancer Metabolism & Systems Toxicology Group, Division of Cancer, Department of Surgery & Cancer, Imperial College London, London, UK. [9]Bradford Institute for Health Research, Bradford Teaching Hospitals NHS Foundation Trust, Bradford, UK. [10]Team of Environmental Epidemiology applied to Reproduction and Respiratory Health, Institute for Advanced Biosciences (IAB), Inserm, CNRS, Université Grenoble Alpes, Grenoble, France. [11]Centre for Research in Epidemiology and Statistics (CRESS), Inserm, Université de Paris, Paris, France. [12]Hospital del Mar Medical Research Institute (IMIM), Barcelona, Spain. [13]Division of Climate and Environmental Health, Norwegian Institute of Public Health, Oslo, Norway. [14]Department of Environmental Sciences, Vytautas Magnus University, Kaunas, Lithuania. [15]Department of Social Medicine, Faculty of Medicine, University of Crete, Heraklion, Greece. [16]Department of Population and Public Health Sciences, University of Southern California, Los Angeles, CA, USA. [17]Computational Biology program, CIMA-University of Navarra, Pamplona, Spain. [18]Medicine Genomics Group, Centro de Investigación Biomédica en Red Enfermedades Raras (CIBERER), University of Santiago de Compostela, CIMUS, Santiago de Compostela, Spain. [19]Galician Foundation of Genomic Medicine, Instituto de Investigación Sanitaria de Santiago de Compostela (IDIS), Servicio Gallego de Salud (SERGAS), Santiago de Compostela, Galicia, Spain. [20]Oncology Safety, Clinical Pharmacology and Safety Sciences, R&D, AstraZeneca, Cambridge, UK. [21]These authors contributed equally: Léa Maitre, Mariona Bustamante, Juan R. González, Hector C. Keun, Martine Vrijheid. ✉e-mail: martine.vrijheid@isglobal.org

