## [Peer Review File · Nature Communications]

Multi-omics signatures of the human early life exposomeREVIEWER COMMENTS

Reviewer #1 (Remarks to the Author):

The authors performed a multi-omics exposome analysis in the Human Early Life Exposome (HELIX) project, a multi-centre cohort of 1,301 mother-child pairs, to characterize how prenatal and childhood exposures associated with molecular profiles. They identify numerous associations, and suggest that epigenome may best capture prenatal exposures while all omic layers appear to be responsive to exposures during childhood. They have created an online resource where these findings should be easily accessible to the broader scientific community (I did have some trouble finding the results when I went to the website - will this be publicly available upon acceptance?). While the results of an exposome, multi-omic analysis would be of interest to the broader scientific community, these analyses feel superficial, partially because the methods are poorly explained and very few actual results are presented, almost everything is located in the extensive supplemental materials. This would benefit from additional analyses to provide stronger evidence that the reported results are robust, and more of the actual results should be reported within the text, with vastly reduced interpretation/discussion.

Minor Issues:

1. Could the authors please clarify why each molecular omic was analyzed separately/independently - I expected a multi-omics analysis to leverage the interrelationships between molecular omics. There is a small section where methylation, mRNA and miRNA interrelationships are considered, but only among subsets of features that were identified with some of the exposures. Provide some additional explanation/justification for this overall approach.
2. The labelling (headings, and x- and y-axes) for the supplemental figures are not intuitive, and made it difficult to link these supplemental materials to the conclusions presented in the main text.
3. Table 1 - the following column labels should be revised to be more clear: "Exposures, ordered by degree", "prot", "Met_s", "met_u".
4. Figures are not very easily interpretable without more detailed captions.

Major Issues:

There are so many results being reported, but they are almost entirely in the supplemental materials. The actual results section has almost no statistics reported within the text and only makes general statements about the numbers of associations, and the tables and figures are not very intuitive or interpretable. Throughout the results section, the authors provide references and citations to compare their results to other findings from the literature. The vast majority of this should go in the discussion section, and additional statistics and data that support these findings should be added to the results section. For instance - what are the genes that are annotated to the CpGs that yielded the strongest associations, or that are impacted by multiple different exposures? The results section on the molybdenum associations, for example, only report that lots of associations with CpGs were observed and that most CpGs were hypomethylated. But no effect sizes were reported, no information about whether these CpGs are correlated with each other, or what genes these are linked to? Instead, there is almost an entire page about molybdenum exposure and health effects... But I don't really understand what associations were actually observed with Mo. Does differential methylation that is associated with maternal Mo persist into childhood? Does it effect the gene expression of nearby genes? are the genes that are impacted by Mo within common genetic regions, or belong to the same biological pathways? While I highlight this issue for the Mo associations, each sub-section within the results suffers from similar issues. Overall, this analysis feels fairly superficial, with lots of interpretation. This would benefit from more thorough and careful analysis and more detailed explanation of their findings, then focusing in on interpretation of the findings in a more focused discussion section. Some additional specific comments and issues are below:

1. There is no methods section in the main text... There are extensive methods in the supplementals,

but most of this needs to be in the main manuscript, and some of this needs to be mentioned in the results (ie. multiple testing correction methods, alpha-thresholds, confounder adjustments) so that the results can be interpreted within the context of the methods being used.

2. Was the maternal Cd analyses performed while adjusting for maternal Smoking? It seems like maternal smoking is an important confounder of the potential relationships between maternal Cd and offspring molecular features. If maternal smoking was not adjusted for, please re-run this model while including this adjustment. And in general, it isn't clear if any exposure-specific confounders were included in the modeling, or if there were any differences in adjustment for the pregnancy-specific and child-specific analyses. This needs to be more clearly explained, because it seems like no exposure-specific confounders were considered, which would mean that many of the results may be biased by residual confounding.

3. Results section; lines 156-165: Clarify how the exposome-omics-wide models are being run (independent and dependent vars), the number of exposure variables being studied, the number of molecular features being studied, and how multiple testing was corrected for. Was there a single multiple-testing correction for the overall ~10mill tests, or were corrections being applied to each -omic analysis separately? This information needs to be clear to the reader when they first engage with the results.

4. The finding that the early life epigenome appears to be most responsive to prenatal exposures is quite interesting - 70% of associations were with DNA methylation. However, since it isn't clear what the proportion of CpGs made up the input of molecular features, it isn't actually clear if this is the case. If 70% of the omic-features being studied are CpGs, then you would expect ~70% of your findings to be with CpGs, even if all molecular features were similarly responsive to these exposures. Please add some clarity here.

5. I am concerned about the p-values that are presented on the miami plots (Figure 2). There are horizontal bands of $-\log_{10}(p)$ values that appear to come from almost identical p-values, which seems unlikely. I would expect to see a lot more variation in the p-values. This issue is most apparent for the childhood exposure results (2D), but is present throughout these plots. Please check these results, and update the plots or explain why there are numerous bands of identical p-values. Also, the x-axis is not described, is there vertical line for each exposure? Please clarify how the data-points are ordered on the x-axis.

6. I'm confused by the network analysis and what the goal or hypothesis was, as well as the interpretation of the results; lines 179-200. I think the authors are demonstrating sets of molecular features that are impacted by multiple different exposures. If so, please state this at the beginning of the results section; if not, then clarify the goal at the beginning of this section. Then the authors performed unsupervised clustering, but it isn't clear what is being clustered on (the coefficients from the Exp-omic-EWAS regression results, or the values of the molecular features themselves? And, what does connectedness mean in this section, aside from correlations between variables? Does this tell us about genes or biological processes that are affected by different exposures, about common sources of exposure? Also, the figure captions add very little to the ability to interpret these, they are practically copied from the main text. The caption plus the figure should stand alone and allow the reader to

7. Lines 497-509 - here the authors do some important sensitivity analyses to examine consistency of the associations within a homogenous ancestry group, and across the different cohorts. But all of the results are in the supplemental materials. This section should be near the top of the results section, and much more detailed, because this is where the authors can provide evidence that their results are in fact robust and not driven by individual cohorts, technical artefacts, or outliers (did the authors investigate why some associations were highly cohort-specific?).

Reviewer #2 (Remarks to the Author):

Summary: This manuscript presents a thorough analysis and discussion of the relationships between >100 environmental exposures and four different omic "layers" (DNA methylation, proteins, miRNA/mRNA, and metabolites). The authors should be highly commended for a thorough analysis and a well-written manuscript that was easy to follow, given the complexity and breadth of the analyses conducted. I think this study is generally well conducted and presents a clear contribution, particularly to the nascent field of exposomics. However, I believe some revision is necessary as there are a few questions and comments that should be addressed, which may affect the interpretation of the results in some areas.

Major Comments/Questions:

- While the authors described and defined "exposome" well, relevant details on the molecular features outcome data is somewhat lacking, especially for the plasma/serum/urine measurements of metabolites and proteins. At the minimum, there should be some description of what was captured by each platform and how many features were examined in relation to the exposome to help place the results in context.
- How much missing data was present and had to be imputed? Why generate 20 imputed datasets if only one was used? Why would one not use all 20? At the very least, can the authors please provide some evidence to reassure that the imputation itself does not meaningfully affect the results?
- Related to above – were there any missing molecular feature data? If so, how were they treated?
- Why did the authors choose to use ENT correction instead of FDR? Correct me if I am wrong, but ENT correction essentially tests whether the independent X is associated with an entire omic, assuming strong correlations across the individual features of each omic. It does not seem suitable for inference on an individual exposure/feature level.
- One major interpretation of the data is that the "methyloome best captures the persistent influence of pregnancy exposures". It is not so clear that this is the only (or even most likely) explanation for the observed findings. For example, the relatively high number of CpG sites may also be a reflection of the number of tests conducted (e.g. ~386,000 CpGs vs. ~60,000 other features combined). While one can see some disparity between pregnancy exposure results and childhood exposure results, the overall design (small capture of proteome and metabolome) and evidence does not necessarily support the assertion, unless additional rationale, arguments, and evidence is provided.
- Related to the above, the authors compare and discuss results from different biological matrices (lines 471-479). Can the authors clarify what they mean by the "exposome" here? Is it defined as having a statistically significant association with at least 1 exposure? Also, it would be helpful to show data on the correlation of these metabolites across biological matrices and discuss its implications.

Minor Comments/Questions:

- In the introduction (lines 101-113), there seems to be some (unintentional) conflation of DNA methylation and epigenetics and DNAm is certainly not the only epigenetic mechanism that affects gene expression. It would be better if the difference between "epigenetics marks" and "DNAm" was more clear and distinct.
- Can the authors please clarify the availability and form of the indoor air pollution data? It is unclear how many people are in the "subgroup" and what "two seasons" refer to.
- Minor typos in the supplement.
- Perhaps I am not looking in the right places, but I could not find the explanation and captions for the supplemental figures and they were not intuitive to me.
- Were the CpG associations with Cadmium consistent with previous studies of the topic?
- Lines 501-502 – what would be considered a "major" difference in effect size?

REVIEWER COMMENTS

PREAMBLE

We thank the editor and the reviewers for their constructive comments. They have helped us to improve the manuscript. We have addressed comments one by one below, but before, we would like to highlight main changes of this updated version of the manuscript:

1. **Methods:** In the previous version of the manuscript, the Method section was in Supplementary Material. Now the manuscript contains a description of the Methods, after the Discussion, as well a brief summary in the Results section to facilitate the comprehension of how we are addressing each question methodologically. An extended description of data acquisition and statistical analyses can be found in Supplemental Information.
2. **Results:** We have reorganized the Results section. Now, the first sections describe the data, the Exposome-omics-Wide Association Study (ExWAS), and the web catalogue. Most importantly, we describe the 1,170 exposure-omics associations that pass multiple testing corrections and show the list of top results in the main text: Tables 1 and 2 with effect sizes and p-value statistics. We then discuss their robustness to ancestry, body mass index, and cohort. Subsequently, we show the multi-omics exposure networks built using the 1,170 associations, and we clarify the aim of such analysis. Then, we describe the results of groups of exposures based on the network clusters, in a more succinct form than the previous version. Finally, we assess if the results are replicated across molecular layers and biological matrices.
3. **Discussion:** We have moved most of the text contextualizing our findings with the literature in terms of health consequences into the Discussion to avoid the impression that we are overinterpreting the findings.
4. **Data:** While the exposome data has previously been analysed in publications relating to health phenotypes, this is the first time we present systematically the associations of the HELIX early life exposome vs. omics signatures for publication. In none of these publications data at the individual level has been made publicly available. This decision responds to policies of data protection within each of the HELIX cohorts. Recognizing the importance of data sharing and in response to these restrictions, we have built the HELIX-exp-omics web catalogue (<https://helixomics.isglobal.org/>), which will be publicly available once the manuscript is published and from where all summarized exposure-omics associations can be downloaded. We also want to highlight that individual level data can be transferred to non-HELIX researchers after signature of a data transfer agreement (DTA), under our established data access policy. All this is quoted at the end of the manuscript in the Data availability section (page 21 – lines 633-638): “The summarized results (exposure, omics biomarker, effect, standard error, p-

value) generated during this study are available at <https://helixomics.isglobal.org/>. The raw data supporting the current study are available from the corresponding author on request subject to ethical and legislative review. The “HELIX Data External Data Request Procedures” are available with the data inventory in this website: <http://www.projecthelix.eu/data-inventory>.”

Here we provide the reviewers and editor the information to access the HELIX-exp-omics web catalogue (user: helix; pass: HELix888OmicS@), which was initially in the cover letter.

Point by point answers:

Reviewer #1 (Remarks to the Author):

In the performed a multi-omics exposome analysis in the Human Early Life Exposome (HELIX) project, a multi-centre cohort of 1,301 mother-child pairs, to characterize how prenatal and childhood exposures associated with molecular profiles. The identify numerous associations, and suggest that epigenome may best capture prenatal exposures while all omic layers appear to be responsive to exposures during childhood. They have created an online resource where these findings should be easily accesible to the broader scientific community (i did have some trouble finding the results when I went to the website - will this be publicly available upon acceptance?). While the results of an exposome, multi-omic analysis would be of interest to the broader scientific community, these analyses feel superficial, partially because the methods are poorly explained and very few actual results are presented, almost everything is located in the extensive supplemental materials. This would benefit from additional analyses to provide stronger evidence that the reported resutls are robust, and more of the actual results should be reported within the text, with vastly reduced interpretation/discussion.

Major Issues:

QUESTION 1: Result reporting, interpretation and discussion

There are so many results being reported, but they are almost entirely in the supplemental materials. The actual results section has almost no statistics reported within the text and only makes general statements about the numbers of associations, and the tables and figures are not very intuitive or interpretable. Throughout the results section, the authors provide references and citations to compare their results to other findings from the literature. The vast majority of this should go in the discussion section, and additional statistics and data that support these findings should be added to the results section. For instance - what are the genes that are annotated to the CpGs that yielded the strongest associations, or that are impacted by multiple different exposures? The results section on the molybdenum associations, for example, only report that lots of associations with CpGs were observed and that most CpGs were hypomethylated. But no effect sizes were reported, no information about whether these CpGs are correlated with each other, or what genes these are linked to? Instead, there is almost an entire page about molybdenum exposure and health effects... But I don't really understand what associations were actually observed with Mo. Does differential methylation that is associated with maternal Mo persist into childhood? Does it effect the gene expression of nearby genes? are the genes that are impacted by Mo within

common genetic regions, or belong to the same biological pathways? While I highlight this issue for the Mo associations, each sub-section within the results suffers from similar issues. Overall, this analysis feels fairly superficial, with lots of interpretation. This would benefit from more thorough and careful analysis and more detailed explanation of their findings, then focusing in on interpretation of the findings in a more focused discussion section.

ANSWER M1: We thank the reviewer for his/her comments. We have decided to split this first main comment in different concrete questions to be able to address them.

1. Too many results listed and without much detail

We agree with the reviewer that the amount of associations we described has impacted negatively on the main message of the manuscript. This is because we wanted to explain the whole HELIX-exp-omics catalogue, which is an impossible task. Thus, and as indicated in the preamble, we have extensively restructured the manuscript with the aim of giving a clearer message of the main findings, and also importantly, of the utilities of the web catalogue.

Moreover, to provide the additional statistics requested by the reviewer, we have included in the main text two tables with “top” exposure-omics associations. “Top” associations are defined as those associations with a p-value $<1E-09$. This threshold represents the nominal p-value divided by all tests performed (explained in detail in Supplemental Information), and it is merely used to make this selection since it was impossible to show all 1170 associations in the main text. These tables show the exposure-omics pairs, the effect size, the p-value of the association and the full gene or metabolite annotation when available (Tables 1 and 2, for the pregnancy and childhood exposome respectively). In the case of CpGs, we show the chromosome, position, gene name and relative gene position. All other exposure-omics associations are listed in Table S2 and in the HELIX-exp-omics web catalogue.

As raised by the reviewer, results on maternal molybdenum (Mo) were missing details. In this updated version of the manuscript we have reported the essential information of the results for molybdenum: number of CpGs (and loci), effect size, annotated genes, pathways and overlap with the EWAS Atlas/Catalog (see Table 1 and text in page 10 – lines 306-312): “Maternal Mo was related to the methylation levels of 72 CpGs in children, representing 63 loci (cluster preg#2). No relevant gene-sets were identified for genes annotated to these 72 CpGs, but 13 of them have previously been related to gestational age (**Figure 6A; Table S5**). The list of genes annotated to top Mo CpGs (p-value $<1E-09$) can be found in **Table 1**. Mo acts as a co-factor of 4 human enzymes which are involved in various key reactions, including the regulation of sulphur-containing amino acids such as methionine⁴². In our dataset, maternal Mo was associated with higher methionine levels in childhood (**Table S2**).

As such, we now provide a better summary of the novel Mo findings. Finally, to answer the question: “Does differential methylation that is associated with maternal Mo persist into childhood?” Methylation was measured only in childhood, therefore we indeed report persistent associations with maternal Mo. We have clarified this in the text (page 13, lines 395-397: “methylation changes in a remarkable number of CpGs, which were persistent at least until childhood (when we detected them).”).

2. Discussion of findings in Results section

We agree with the reviewer that the comparison of our results with the literature in the Results section was confusing for the readers. Because of this, we have moved most of these parts to the Discussion. However, as indicated in the preamble we think that keeping a brief summary of results together with some interpretation in the Results section might facilitate the reader to get a message of each part. We have dedicated the final Discussion to sum up and comment up general points of the study: main findings, applications of the catalogue, key variables in the design of exposome-omics association studies, strengths and limitations.

Following with the example of maternal molybdenum (Mo), we have now moved most of the discussion we presented in Results, to the Discussion section (page 13 - lines 392-398): “Furthermore, data generated in this study provide a resource for the development of epigenetic biomarkers of past exposures¹⁸. For instance, it was generally believed that the essential element molybdenum (Mo) is safe for human health⁵⁷, however there is growing evidence that excess of Mo is associated with developmental effects and with adverse health outcomes^{24,58–62}. In this study, maternal levels of Mo were associated with methylation changes in a remarkable number of CpGs, which were persistent at least until childhood (when we detected them). The methylation in these CpGs could be used to predict prenatal exposure levels“.

3. Superficial analyses

Our study aimed to systematically document all associations between the exposome and the molecular phenotypes. It forms a basis for future in-depth studies into the specific associations. Because of the scope of this study, it was necessary to provide summaries of our findings in the main text, for example through Miami plots and network analyses, and refer to supplemental material and the online catalogue for detailed results. Nevertheless, our workflow included analyses of robustness and deeper analyses. In order to clarify to the reader these efforts, we have restructured and clarified the results:

- 1) We conducted three sets of sensitivity analyses to evaluate robustness of the findings (See QUESTION M9: Sensitivity analyses). We have moved the analyses of “Robustness of results with respect to ancestry, child zBMI and cohort” upfront, page 7, after the first results paragraph. We added Figure 3 Robustness of main exposome-omics associations to the main text.
- 2) In some cases, we performed deeper analyses to illustrate some of the challenges of the exposome studies. For instance, we addressed the correlation between exposures, by separating the effects of cadmium (Cd) on DNA methylation in: cadmium-specific effects, and shared effects with tobacco smoke (See further QUESTION M3: Maternal tobacco smoking and maternal cadmium (Cd)).
- 3) Biological interpretation of the findings was done using diverse strategies including systematic literature overlap and functional enrichment analyses with several public databases.
- 4) Reproducibility of the findings was investigated across exposure periods (pregnancy and childhood), biofluids (12 urine/serum metabolites) and molecular layers with the aim of giving some guidance to future exposome studies

Our new workflow figure 1, highlight all these additional steps to better guide the readers.

QUESTION M2: Methods

There is no methods section in the main text... There are extensive methods in the supplementals, but most of this needs to be in the main manuscript, and some of this needs to be mentioned in the results (ie. multiple testing correction methods, alpha-thresholds, confounder adjustments) so that the results can be interpreted within the context of the methods being used.

ANSWER M2: We have added a Methods section in the main text of the new version of the manuscript after the discussion. In addition, we have added, throughout the manuscript, summaries of the methods applied to facilitate the comprehension of what was done in each Results section.

To specifically answer the reviewer about the multiple testing correction method we have added this text (pages 6 – lines 163-169): “To identify statistically significant exposure-omics associations, correction for multiple comparisons was applied for each exposure within each omics dataset. For this, we considered significant associations the ones with p-values below a False Discovery Rate (FDR) of 0.05 for genome-wide omics, and below a modified version of the Bonferroni cut-off for the proteins and metabolites (which consists in dividing the nominal p-value by the effective number of tests (ENT) determined from the correlation structure of the omics dataset, see Table S1G; Supplemental Information)”.

QUESTION M3: Maternal tobacco smoking and maternal cadmium (Cd)

Was the maternal Cd analyses performed while adjusting for maternal Smoking? It seems like maternal smoking is an important confounder of the potential relationships between maternal Cd and offspring molecular features. If maternal smoking was not adjusted for, please re-run this model while including this adjustment.

ANSWER M3: The main ExWAS models were not mutually adjusted for other exposures (thus maternal smoking was not adjusted for maternal Cd, nor vice versa). However, as we had the same concern as the reviewer when preparing the manuscript, we also ran the analyses of maternal Cd and DNA methylation in the strata of non-smoker mothers, which is similar to adjusting for maternal tobacco smoking. This was explained in the Result section (page 9 – lines 255-262): “Prenatal cadmium (Cd), a heavy metal, was associated with child blood methylation, and clustered with maternal smoking in cluster preg#1. This could be partially explained by the fact that Cd is a component of tobacco³⁷ and in our dataset mothers who smoked showed almost twice the level of Cd compared to non-smokers. However, we identified 14 additional CpGs that were unique to Cd (**Figure 2A-2B; Table S7A**). When restricting our analysis of maternal Cd to non-smoker mothers (N=998), 51 CpGs (48 loci) were identified (**Figure S2C-2D; Table S7B**). These did not overlap with known smoking effects, nor with CpGs associated with urinary Cd in adult blood or with placental Cd in placental tissue^{38,39}.”

QUESTION M4: Confounders

And in general, it isn't clear if any exposure-specific confounders were included in the modeling, or if there were any differences in adjustment for the pregnancy-specific and child-specific analyses. This needs to be more clearly explained, because it seems like no exposure-specific confounders were considered, which would mean that many of the results may be biased by residual confounding.

ANSWER M4: The main ExWAS models were adjusted for cohort, child's age, sex, child BMI, ancestry, maternal education and omics specific technical covariates. In addition, we explored through sensitivity analyses, three main sources of variability in the exposome and omics dataset, which are child's ancestry, child's BMI and cohort stratification. This is explained in more detail in QUESTION M9. We believe that by adding these main covariates we accounted for the main sources of confounding. However, we agree with the reviewer that some residual confounding might still be present. This has been commented in the Discussion together with the need of replication and of other study designs for causal inference (page 17 – lines 503-506): “Finally, although our models were adjusted for confounders, residual confounding might still be present and causal links would need to be proven through interventions, Mendelian randomization analyses, cross-contextual studies or in vivo / in vitro models in order to move to therapeutic and preventative strategies.”

Exposure specific confounders were considered only to disentangle the relationship between maternal tobacco smoking during pregnancy and maternal cadmium (Cd), but not for other exposures (see QUESTION M3). We have added this limitation in the Discussion (page 17 – line 493-499): “Moreover, the correlated nature of the exposome makes identification of driving exposures difficult. Here we tried to separate the effect of maternal tobacco smoking and maternal cadmium (Cd) by running stratified models and identified Cd-specific effects. Besides tobacco smoke, Cd might have other origins such as rice, potatoes and wheat, when frequently consumed in large quantities⁹³. Mixture or multi-pollutant approaches aim to tackle this more systematically, however these are not yet suitable for high-dimensional omics datasets such as ours^{94,95}.”

QUESTION M5: Methods of the ExWAS

Results section; lines 156-165: Clarify how the exposome-omics-wide models are being run (independent and dependent vars), the number of exposure variables being studied, the number of molecular features being studied, and how multiple testing was corrected for. Was there a single multiple-testing correction for the overall ~10mill tests, or were corrections being applied to each -omic analysis separately? This information needs to be clear to the reader when they first engage with the results.

ANSWER M5: We thank the reviewer for this comment. The models of the Exposome-omics-Wide Association Study (ExWAS) were run independently, meaning that each exposure was tested vs each molecular feature adjusting for covariates. We have modified this paragraph in the Results section (page 5 – line 152-162) to clarify this point:

“Results of the Exposome-omics-Wide Association Study (ExWAS)

We first systematically tested the association between each exposure variable and each molecular feature, successively and independently, through an Exposome-omics-Wide Association Study (ExWAS), using an analogous statistical approach to that of

Genome-Wide Association Studies (GWAS) (**Figure 1D; Supplemental Information**). Overall, we tested >30 M exposure-omics associations (>0.3 M molecular features * ~100 exposures * 2 exposure periods) through linear regression models adjusted for the same set of confounders: cohort, child's age, sex, z-score body mass index (z-BMI), ancestry, maternal education and omics specific covariates. Results of all these associations can be viewed in the web catalogue: <https://helixomics.isglobal.org/> (for genome-wide omics platforms only results with p-values <0.01 are included). ”

We have clarified our multiple testing approach in the Results section (page 6 – lines 161-167): “To identify statistically significant exposure-omics associations, we corrected for multiple testing by accounting for multiple omics features within each omics dataset separately. For the genome-wide omics (DNA methylation, gene expression, miRNA) we considered as statistically significant associations those with a p-value below a False Discovery Rate (FDR) of 0.05. For the proteins and metabolites we used a modified version of the Bonferroni cut-off, which consists of dividing the nominal p-value by the effective number of tests (ENT) determined from the correlation structure of the omics dataset, as shown in **Table S1G** (see also **Supplemental Information**).” We also clarified this in the Methods, and Supplemental Information. In Table S1G, we are showing the effective number of tests and p-value thresholds for each omics and for the overall number of tests performed.

QUESTION M6: The epigenome best captures the pregnancy exposome

The finding that the early life epigenome appears to be most responsive to prenatal exposures is quite interesting - 70% of associations were with DNA methylation. However, since it isn't clear what the proportion of CpGs made up the input of molecular features, it isn't actually clear if this is the case. If 70% of the omic-features being studied are CpGs, then you would expect ~70% of your findings to be with CpGs, even if all molecular features were similarly responsive to these exposures. Please add some clarity here.

ANSWER M6: We agree with the reviewer that this statement “The epigenome best captures the pregnancy exposome” needed some revision, given that the CpGs represent 87% of all molecular features. We have therefore decided to eliminate this statement from the Abstract and the Results section and comment on this issue in the Discussion (page 15 – lines 463 - 461): “Our results indicate that the choice of molecular layer and biological matrix is key in the design of exposome studies. We found that the prenatal environment was mainly associated with the methylome showing sustained changes until childhood. This goes in line with previous publications that suggest that the epigenome acts as the main source of cellular ‘memory’ and plasticity. In contrast, recent exposures during childhood were associated with features across all omics layers. Evidence to date suggests that the metabolome in particular is strongly influenced by the immediate environment, and may thus be more sensitive for detecting associations in cross-sectional settings²¹. Nevertheless, many cross-sectional associations with the methylome were found and, although fewer, long-term associations with other omics were also found.”

References:

Cavalli, G. & Heard, E. Advances in epigenetics link genetics to the environment and disease. *Nature* (2019)

Tsai et al. Smoking induces coordinated DNA methylation and gene expression changes in adipose tissue with consequences for metabolic health. *Clinical Epigenetics* (2018)

QUESTION M7: P-values in Miami plots

I am concerned about the p-values that are presented on the miami plots (Figure 2). There are horizontal bands of $-\log_{10}(p)$ values that appear to come from almost identical p-values, which seems unlikely. I would expect to see a lot more variation in the p-values. This issue is most apparent for the childhood exposure results (2D), but is present throughout these plots. Please check these results, and update the plots or explain why there are numerous bands of identical p-values. Also, the x-axis is not described, is there vertical line for each exposure? Please clarify how the data-points are ordered on the x-axis.

ANSWER M7: We re-created the Miami plots for Figure 2 so the p-value variation is visible and there are no more identical p-values as horizontal lines. Each vertical line represents one exposure however some jitter (adding random noise to data in order to prevent overplotting) on the x-axis was added so the p-values with similar values don't completely overlap. We have clarified the headings of this Figure.

QUESTION M8: Network analyses

I'm confused by the network analysis and what the goal or hypothesis was, as well as the interpretation of the results; lines 179-200. I think the authors are demonstrating sets of molecular features that are impacted by multiple different exposures. If so, please state this at the beginning of the results section; if not, then clarify the goal at the beginning of this section. Then the authors performed unsupervised clustering, but it isn't clear what is being clustered on (the coefficients from the Exp-omic-EWAS regression results, or the values of the molecular features themselves? And, what does connectedness mean in this section, aside from correlations between variables? Does this tell us about genes or biological processes that are affected by different exposures, about common sources of exposure? Also, the figure captions add very little to the ability to interpret these, they are practically copied from the main text. The caption plus the figure should stand alone and allow the reader to

ANSWER M8:

We clarified below the goal of building the period-specific and multi-omics networks, and how we created and interpreted them.

Methodology: To construct the networks, we separated the 1,170 exposome-omics associations into 249 for the pregnancy network and 921 for the childhood network. Because of this we call it a period-specific networks. In each network we considered as nodes all the exposures and all the features from different molecular layers (52 and 209 unique exposures and molecular features for the pregnancy; and 84 and 454 for the childhood network). Because of this we call it a multi-omics network. Each node type (exposure family, CpG, protein, etc.) was indicated with a different shape and/or color. Then, using Cytoscape 3.6.1 (<http://cytoscape.org>) edges (connections) were drawn between them (which indeed are the 1,170 associations). If an exposure was connected to more than one molecular feature, it had several edges. If a molecular feature was connected to more than one exposure, it had several edges. In this way, we can visually

identify which are the exposures and molecular features that are highly connected. Moreover, for each node we calculated several parameters that measure connectivity and which are reported in Tables S6A-S6B. Finally, using the Community Clustering (GLay) we identified clusters, which are defined as densely connected regions in the network. Connectedness means that an exposure is associated (given the p-value threshold we have defined) with several molecular features.

We have re-written the Methods and Supplemental Information to describe how these networks were created (similar text as the answer given to the reviewer).

Page 20 – lines 580-598: “We conducted network analyses using the list of ExWAS associations passing multiple testing correction (N=1,170). We built a network for each exposure period (period-specific), and each network contained all the molecular layers (multi-omics). Molecular features and exposures were considered as the nodes of the network and the edges represented omics-exposure associations (based on the ExWAS results). Networks visualization was carried out using Cytoscape 3.6.1 (<http://cytoscape.org>) and was automatically arranged using the Cytoscape force-directed layout, which aims to highlight the underlying topology of the graph. The association effect size was set as the numeric edge column to use as a weight for the length of the edges. Clustering of the childhood network was done based on Community Clustering (GLay) in order to find densely connected regions in the network.”

Goal: We use the networks to identify which exposures and molecular features were inter-connected. In other words, and as the reviewer says, we try “to identify sets of molecular features that are impacted by multiple different exposures”. We have clarified the goal of the network analyses is in the text:

Page 7 – lines 211-215: “To visualize whether a molecular feature was connected to several exposures, and vice versa, we built period-specific multi-omics networks, using the 1,170 significant exposome-omics associations. The nodes of these networks are the 538 unique molecular features or exposure factors involved in these associations, and the edges are the 1,170 associations identified between them (Supplemental Methods).”

Given that connectedness is important to describe the network, we are now reporting the values of two of these parameters in the results section, degree and shortest path:

Page 7 – lines 216-219: “The pregnancy exposome network, mostly composed of CpGs (70%), was very disconnected having on average 1.3 connections per node (i.e. degree) and an average shortest path length of 1.9 (Figure 4, Table S4A). This number represents the average length (number of nodes) of the shortest path between each node and any other node, 1.9 being a low value.”, and page 8, lines 225-227 “The childhood exposome network was more densely connected, with an average of 1.9 connections per node and an average shortest path length of 4.3. The biggest connected component contained 90% of all nodes (Figure 5).”

Biological interpretation: The clusters per se do not inform about the biological process that are affected by different exposures or about the source of exposures. This comes from our biological interpretation of the clusters using information from public databases. To avoid confusion and to highlight the extra step we conducted to interpret the clusters, we have decided to eliminate this sentence from the Results section: “The clusters reveal both potential biological pathways and potential routes or sources of

exposure, as further described below”. Also the title of the section has been simplified to “Network integration of multi-omics signatures of the exposome”.

Figure caption: We clarified the captions of the Figures 4 and 5 (pregnancy and childhood networks). See text for Figure 5, as an example

“Network map of the multi-omics signatures of the childhood exposome.

Network visualization of the childhood Exposome-omics-Wide Association Study (ExWAS). Nodes of the network are depicted with a different colour/shape depending if they are exposures or features of a particular molecular layer (see legend). An exposure and a molecular feature were connected with an edge if their association was statistically significant (blue if positively and red if inversely associated). Only connected components with at least two molecular features were displayed. An exposure and a molecular feature were connected if their association was statistically significant and only connected components with at least two molecular features were displayed. The childhood exposome network was diverse in terms of omics features represented and the level of interconnection, with the biggest connected component containing 90% of all nodes. Within this network, 13 clusters were identified using an unsupervised structural cluster analysis (see **Supplemental Information**), and were annotated in the figure. The summary table with the cluster characteristics are in Table 3 and a full table with the node attributes can be found in **Table S4B**. “

QUESTION M9: Sensitivity analyses

Lines 497-509 - here the authors do some important sensitivity analyses to examine consistency of the associations within a homogenous ancestry group, and across the different cohorts. But all of the results are in the supplemental materials. This section should be near the top of the results section, and much more detailed, because this is where the authors can provide evidence that their results are in fact robust and not driven by individual cohorts, technical artefacts, or outliers (did the authors investigate why some associations were highly cohort-specific?).

ANSWER M9: As suggested by the reviewer, we have moved the sensitivity analyses just after the description of the ExWAS at the beginning of the Results section, and presented the comparison of the 1,170 exposure-omics associations between main and sensitivity models in Figure 3 (new figure). The section is divided in three parts:

1. Ancestry: We compare the model adjusted for ancestry vs. the model in only European ancestry children. No major differences were found as seen in the scatter plot comparing the effect sizes of the two models (Figure 3A).
2. Z-score body mass index (z-BMI): We compared the model adjusted for z-BMI vs. the model unadjusted for z-BMI. In this comparison we saw differences larger than doubling in effect size for some proteins produced by the adipose tissue (IL1beta, leptin and IL6) and lipophilic chemical pollutants (PCB170, PCB153 and PCB180) (Figure 3B). But for the rest of exposures, z-BMI adjustment did not affect the associations.
3. Cohort: We repeated the ExWAS within each one of the cohorts. Then, for the significant 1,170 exposure-omics associations we conducted a fixed and random-effects inverse variance weighted meta-analyses, calculated heterogeneity (I^2) and did forest plots. I^2 is reported in the text and in Figure S1 as guidance, but we do not

base our entire interpretation of heterogeneity on this statistic, as it has been seen that it can be biased in meta-analyses with small numbers of studies (von Hippel et al). In contrast, we prefer to show some examples of forest plots of exposure-omics associations with low and high heterogeneity (Figure 3C). We currently do not provide all 1,170 forest plots, due to the large volume of material this would add, but these could be made available as supplemental material if the editor and reviewers recommend this. We added the following description in results:

Lines 201-208 page 7 “we investigated heterogeneity across cohorts by running the 1,170 exposure-omics associations by cohort. Around half of all associations presented heterogeneity values (I^2) <0.5 , with variations by period and molecular layer (**Figure S1**). Besides the I^2 statistic which might be overestimated in meta-analysis with a small number of studies²⁷, we also visually inspected the forest plots. While some associations seemed to be very consistent between cohorts even with a high I^2 (e.g. maternal cadmium and methylation at CpG cg19089201), for others there was more heterogeneity with some cohorts acting as outliers (e.g. child meteorological conditions and serotonin) (**Figure 3C**).”

References:

von Hippel et al. The heterogeneity statistic I^2 can be biased in small meta-analyses. BMC Medical Research Methodology (2015)

Minor Issues:

QUESTION m1: Multivariate analyses

Could the authors please clarify why each molecular omic was analyzed separately/independently - I expected a multi-omics analysis to leverage the inter-relationships between molecular omics. There is a small section where methylation, mRNA and miRNA interrelationships are considered, but only among subsets of features that were identified with some of the exposures. Provide some additional explanation/justification for this overall approach.

ANSWER m1: The objective of this study was to provide a catalogue of omics signatures of the early life exposome. Because of this, we conducted an Exposome-omics-Wide Association Study (ExWAS), thus we tested through linear regression models the association between each exposure vs each molecular feature and corrected the p-value for multiple comparisons. This approach allows:

1. The possibility to adjust for confounders.
2. The ease of interpretation of linear regression models with categorical and continuous predictors.
3. To allow reporting of individual exposure-omics feature relationship for comparison with other studies in the environmental epidemiology field.

The downstream analyses of the 1,170 associations leveraged the interrelationship between molecular omics in several ways:

1. Integrative period-specific network and cluster analyses across all omics were used to visualise common exposures associated with different omics layers.

2. Functional enrichment analyses were performed across four omics layers. Indeed, molecular layers with features which could be easily annotated at the gene level: DNA methylation, gene and miRNA transcription, and proteins were analysed together.
3. Overlap between exposure associations across urine and serum metabolites were tested.
4. Expression quantitative trait methylation (eQTM) and miRNA gene target prediction were used to test methylation, mRNA and miRNA interrelationships.

Alternatives such as multivariate approaches including partial least square (PLS) or canonical correlation analyses (CCA) do not directly allow for adjustment for confounders and interpretation of the biology behind the findings is more complex. Because of this, we discarded them in this first paper of the molecular signature of the early life exposome. They will be explored in the future.

Overall, inter-relationship between omics was tested, in relation with the main objectives of our study, i.e. among features related to the exposome. Other articles on the HELIX omics datasets and omics inter-relationship, excluding the exposome, can be found elsewhere:

1. Expression vs DNA methylation (under review in eLife).
2. SNPs vs. urinary metabolites (Calvo-Serra et al. HMG 2020).

QUESTION m2: Labelling of supplemental figures

The labelling (headings, and x- and y-axes) for the supplemental figures are not intuitive, and made it difficult to link these supplemental materials to the conclusions presented in the main text.

ANSWER m2: Captions of Supplemental Figures can be found in Supplemental Information, together with the Supplemental Figures. We clarified them as suggested by the reviewers.

QUESTION m3: Notation in Table 1

Table1 - the following column labels should be revised to be more clear: "Exposures, ordered by degree", "prot", "Met_s", "met_u".

ANSWER m3: The labeling of the Table 1 (now Table 3) was changed following the reviewer recommendation:

Clusters	Exposures ^a	Total	DNA methylation (CpGs)	Gene expression	miRNA	Proteins	Serum metab.	Urine metab.	Total annotated genes ^b
----------	------------------------	-------	------------------------	-----------------	-------	----------	--------------	--------------	------------------------------------

^aOrdered based on their number of omics associations, from the exposures with the most associations to the ones the most isolated

^bacross CpGs, transcript clusters, miRNA and proteins

QUESTION m4: Figure captions

Figures are not very easily interpretable without more detailed captions.

ANSWER m4: We have reviewed the captions of all figures to clarify them.

Reviewer #2 (Remarks to the Author):

Summary: This manuscript presents a thorough analysis and discussion of the relationships between >100 environmental exposures and four different omic “layers” (DNA methylation, proteins, miRNA/mRNA, and metabolites). The authors should be highly commended for a thorough analysis and a well-written manuscript that was easy to follow, given the complexity and breadth of the analyses conducted. I think this study is generally well conducted and presents a clear contribution, particularly to the nascent field of exposomics. However, I believe some revision is necessary as there are a few questions and comments that should be addressed, which may affect the interpretation of the results in some areas.

Major Comments/Questions:

QUESTION M1: Lack of information on omics data

While the authors described and defined “exposome” well, relevant details on the molecular features outcome data is somewhat lacking, especially for the plasma/serum/urine measurements of metabolites and proteins. At the minimum, there should be some description of what was captured by each platform and how many features were examined in relation to the exposome to help place the results in context.

ANSWER 1: We thank the reviewer for highlighting this. As also suggested by the reviewer #1, we have included a Methods section in the main text (which we missed by error in the previous version). The number of omics variables included in each dataset is specified in Figure 1C and now is also in Table S1C of the Methods section. Moreover, we also have indicated this in the text as follow (page 5 – lines 135): “For these same children, aged between 6-11 years, we performed in-depth multi-omics molecular phenotyping, including measurement of blood DNA methylation (450K, Illumina), blood gene expression (HTA v2.0, Affymetrix), blood miRNA expression (SurePrint Human miRNA rel 21, Agilent), plasma proteins (3 Luminex multiplex assays), serum metabolites (targeted LC-MS/MS metabolomic assay, Biocrates AbsoluteIDQ p180 kit), and urinary metabolites (¹H nuclear magnetic resonance (NMR) spectroscopy) (**Figure 1C; Table S1C**). While blood DNA methylation and transcriptomics were measured genome-wide, the other omics followed a semi-targeted or targeted approach. Plasma proteins included a total of 36 cytokines, apolipoproteins and adipokines (**Table S1D**)²⁵. The serum metabolites (N=177) included amino acids, biogenic amines, acylcarnitines, glycerophospholipids, sphingolipids and sum of hexoses, covering a wide range of analytes and metabolic pathways in one targeted assay (**Table S1E**)²⁶. Urine metabolites (N=44) mainly included amino acids, organic acids, nicotinamides, amines

and gut microbial-derived phenols (**Table S1F**)²⁶.” Finally, we also added in Supplemental Tables S1D-S1F the full list of proteins and metabolites measured in serum and urine, and Supplemental Information explains in detail the omics data acquisition process.

QUESTION M2: Missing values in the exposome

How much missing data was present and had to be imputed? Why generate 20 imputed datasets if only one was used? Why would one not use all 20? At the very least, can the authors please provide some evidence to reassure that the imputation itself does not meaningfully affect the results?

ANSWER M2: Missing values, ranged from no missing values for some child phthalate metabolites to 65% for fast-food intake during pregnancy. The median percentage of missing values per exposure was 4.8% (first quartile 0.9% and third quartile 19.4%). None of the participants had complete data on all exposures. Yet, for 83% of individuals < 30% of exposure variables had missing values. Exposures with more than 30% missing were exclusively pregnancy exposures. A detailed description of missingness in the HELIX dataset has previously been published (Tamayo-Uria et al 2019) and this reference is mentioned in the manuscript.

For the HELIX exposome-health outcome studies, 20 imputed datasets were generated using a chained equations method (White et al. 2011) implemented in the mice R package (Buuren and Groothuis-Oudshoorn et al. 2011). Results of each dataset were combined using the Rubin’s rule that incorporates the variability due to uncertainty in imputations to the final confidence intervals and p-values.

However, in the HELIX exposome-omics analyses (this manuscript) we only used one of the 20 imputed datasets. We decided to use only one imputed dataset due to time computing restrictions. For this manuscript we have run >180 M linear regressions (100 exposures * 2 periods * 0.3 M molecular features * 3 models (main and 2 extra sensitivity models)). Repeating this for 19 extra imputed datasets would have increased the computational time considerably. We still believe that we made an improvement over previous multi-omics studies by using an imputed dataset and thus correcting for missingness bias.

We have eliminated the sentence about 20 imputed datasets from the Methods section to avoid confusion and now text is as follows (page 18– lines 532-533): “Missing data were imputed using a chained equations method¹⁰⁰ implemented in the mice R package¹⁰¹. One imputed dataset was used in this study.

References:

Buuren, S. van, and Groothuis-Oudshoorn, K. mice : Multivariate Imputation by Chained Equations in R. J. Stat. Softw. (2011)

Tamayo-Uria et al. The early-life exposome: Description and patterns in six European countries. Environ. Int.(2019)

White, I.R., Royston, P., and Wood, A.M. Multiple imputation using chained equations: Issues and guidance for practice. Stat. Med. (2011)

QUESTION M3: Missing values in the omics data

Related to above – were there any missing molecular feature data? If so, how were they treated?

ANSWER M3: Each molecular layer had a different sample size, as seen in Table S1C, because (i) we could not collect exactly the same samples for each biological matrix; and because (ii) some samples were eliminated due to their overall low quality in each omics platform. For instance, in the proteins dataset, we eliminated samples that had ≥ 10 proteins out of the linear range of quantification, and in the methylation dataset we eliminated samples that had discordant sex (comparing sex from methylation data and reported sex). This is explained in detail in Supplemental Information.

Then, within each omics dataset there was a different percentage of missing values, and we have addressed this issue in different ways, following the advice of the experts of each omics. For DNA methylation, gene and miRNA expression, we did not have any missingness (background noise value given for very low signals). For proteins there were some values outside of the limits of quantification. These values were imputed using a truncated normal distribution implemented in the `truncdist` R package (Nadarajah, S., and Kotz, S. 2006). For the metabolites in urine and serum, values given by the platform were kept, even when they were under the limit of detection.

References:

Nadarajah, S., and Kotz, S. The Exponentiated Type Distributions. *Acta Appl. Math.* (2006)

QUESTION 4: Effective number of tests (ENT)

Why did the authors choose to use ENT correction instead of FDR? Correct me if I am wrong, but ENT correction essentially tests whether the independent X is associated with an entire omic, assuming strong correlations across the individual features of each omic.

ANSWER 4: The effective number of tests (ENT) method was preferred for the omics with a small number of features (proteins and serum and urine metabolites) because:

1. False Discovery Rate (FDR) requires a large number of tests for the calculation of the null distribution of p-values, which is not available for omics with small numbers (N between 36 and 177).
2. The high correlation among features in these omics makes the standard Bonferroni correction too restrictive.

Because of this we applied a modified version of the Bonferroni correction, which takes into account the correlation among features by estimating a number of effective tests (ENT), for which we corrected the nominal p-value. This has been explained in the Results section (page 6 – lines 161-167): “To identify statistically significant exposure-omics associations, correction for multiple comparisons was applied for each exposure within each omics dataset. For this, we considered significant associations the ones with p-values below a False Discovery Rate (FDR) of 0.05 for genome-wide omics, and below a modified version of the Bonferroni cut-off for the proteins and metabolites

(which consists in dividing the nominal p-value by the effective number of tests (ENT) determined from the correlation structure of the omics dataset, see **Table S1G; Supplemental Information**).” More information can be found in Methods and Supplementary Information.

QUESTION 5: The methylome best captures the persistent influence of pregnancy exposures

One major interpretation of the data is that the “methylome best captures the persistent influence of pregnancy exposures”. It is not so clear that this is the only (or even most likely) explanation for the observed findings. For example, the relatively high number of CpG sites may also be a reflection of the number of tests conducted (e.g. ~386,000 CpGs vs. ~60,000 other features combined). While one can see some disparity between pregnancy exposure results and childhood exposure results, the overall design (small capture of proteome and metabolome) and evidence does not necessarily support the assertion, unless additional rationale, arguments, and evidence is provided.

ANSWER 5: See ANSWER M6 to first reviewer.

QUESTION 6: Comparison between biological matrices

Related to the above, the authors compare and discuss results from different biological matrices (lines 471-479). Can the authors clarify what they mean by the “exposome” here? Is it defined as having a statistically significant association with at least 1 exposure? Also, it would be helpful to show data on the correlation of these metabolites across biological matrices and discuss its implications.

ANSWER 6: The term “childhood exposome” here refers to any of the 100 exposures measured during childhood. For clarity we have changed the text (page 12 – lines 366-367): “At nominal significance, 27.3% of the urine associations replicated in serum; and 7% of the serum associations replicated in urine (**Tables S9B-S9C**).”

Moreover, as suggested with the reviewer we have added the correlations between the common urinary and serum metabolites in Tables TS4A. These correlations range from 0.001 to 0.39. We have added this text “Not surprisingly, replicated associations involved metabolites with the highest correlation between matrices (carnitine, glycine and creatinine) (**Table S9A**).”

In the discussion we added this page 15:

“Our findings also indicate the importance of the biological matrix. Although we could not make a comprehensive comparison of the urinary and sera metabolomes because of the use of different platforms to assess them, among comparable metabolites, only a few showed consistent associations with the exposome in both biological matrices. Thus, both biological matrices and others should ideally be explored in exposome studies, providing complementary information.”

Minor Comments/Questions:

QUESTION m1: Epigenome vs DNA methylome

In the introduction (lines 101-113), there seems to be some (unintentional) conflation of DNA methylation and epigenetics and DNAm is certainly not the only epigenetic

mechanism that affects gene expression. It would be better if the difference between “epigenetics marks” and “DNAm” was more clear and distinct.

ANSWER m1: We have modified the text as follows page 3, lines 100-105: “It is thought that the epigenome orchestrates cellular responses to environmental perturbations and provides cell memory and plasticity¹⁵. Among all epigenetic marks, DNA methylation is the most studied in epidemiological settings; and among all exposures, tobacco smoke is the most investigated¹⁶⁻¹⁹. To a lesser extent, other diverse exposures, from metals and air pollution to socio-economic factors, have been linked to differential methylation and are catalogued in public databases¹⁶ (<http://www.ewascatalog.org/>).”

QUESTION m2: Indoor air pollution data

Can the authors please clarify the availability and form of the indoor air pollution data? It is unclear how many people are in the “subgroup” and what “two seasons” refer to.

ANSWER m2: Information about how exposure to indoor air pollution was calculated has been added in Supplemental Information (page 3): “Indoor air concentrations of nitrogen dioxide (NO₂), particulate matter <2.5µm (PM_{2.5}), particulate matter absorbance (PMAbs), benzene, and toluene, ethylbenzene, xylene (BTEX) were estimated through a prediction model that combined measurements in the homes of a subgroup of children with questionnaire data from the subcohort. Measurements of indoor NO₂, benzene and TEX were conducted in the homes of **157 participants** as part of the child panel study, which was nested within the HELIX subcohort in all cohorts except MoBa. Participants in the child panel study were followed for **one week in two periods (approximately 6 months apart)**, and the last day of the first week coincided with the subcohort examination, including the completion of the main HELIX questionnaire. NO₂, benzene and TEX sampling lasted 7 days, and PM_{2.5} and PMAbs sampling lasted 24 hours.”.

QUESTION m3: Minor typos in the supplement.

ANSWER m3: We have corrected the errors.

QUESTION m4: Captions of Supplemental Figures

Perhaps I am not looking in the right places, but I could not find the explanation and captions for the supplemental figures and they were not intuitive to me.

ANSWER m4: Captions of Supplemental Figures can be found in Supplemental Information, together with the Supplemental Figures. We have clarified these as suggested by both reviewers.

QUESTION m5: CpGs for cadmium

Were the CpG associations with Cadmium consistent with previous studies of the topic?

ANSWER m5: As indicated in the manuscript, CpG associations with cadmium (Cd) were not consistent with the findings of a previous EWAS of maternal cadmium Cd

measured in placenta and placental DNA methylation: “However, we identified 14 additional CpGs that were unique to Cd; these did not overlap with known smoking effects, nor with CpGs associated with maternal Cd in the placenta (Figure S4A-4B; Table S9A)³⁴.”

Since we submitted the first version of the manuscript, there has been a new publication on Cd and blood DNA methylation in adults (Domingo-Relloso et al.). The authors assessed the association between urinary Cd and self-reported smoking (current-former vs never) with blood DNA methylation in >2000 adults. Models for Cd were adjusted for several covariates as well as smoking status. After multiple testing correction, 6 CpGs were associated with Cd and 4 of them remained significant when the analysis was conducted only in non-smokers. The mediation analyses suggested that Cd is a partial mediator of the association between smoking and differential DNA methylation at specific sites.

We compared the findings by Domingo-Relloso et al. with ours:

1. First, we compared the effect size and statistical significance of 16 CpGs associated with Cd with a FDR <0.25 in Domingo-Relloso et al. with HELIX findings. Seven out of 16 CpGs were available in HELIX and only one of them, cg05575921 in *AHRR*, was inversely associated with Cd in the same direction in the two studies. This CpG was also associated with maternal tobacco smoking.
2. Then, we took the 51 CpGs associated, after multiple testing correction, with maternal Cd in non-smoker mothers in HELIX and searched for their association in Domingo-Relloso et al. Forty-five of them were available in Domingo et al. and 4 of them were nominally significant, but only one had the same direction of the effect in the two studies.

Overall we could not replicate the findings by Domingo-Relloso et al., neither their findings replicated ours. This could, in part, be explained by the differences between the two studies: (i) we assessed maternal Cd in pregnancy instead and Domingo-Relloso et al. own exposure to Cd; (ii) we measured Cd in blood and Domingo-Relloso et al. Cd in urine; (iii) we used the 450K array and Domingo-Relloso et al. the EPIC array.

This has been added in the manuscript (page 9 - lines 256-259): “When restricting our analysis of maternal Cd to non-smoker mothers (N=998), 51 CpGs (48 loci) were identified (**Figure S4C-4D; Table S6B**). These did not overlap with known smoking effects, nor with CpGs associated with urinary Cd in adult blood or with placental Cd in placental tissue^{36,37}.”

References:

Domingo-Relloso et al. Cadmium, smoking and human blood DNA methylation profiles in adults from the Strong Heart Study. EHP. (2020)

Everson TM et al. Cadmium-Associated Differential Methylation throughout the Placental Genome: Epigenome-Wide Association Study of Two U.S. Birth Cohorts. EHP. (2018)

QUESTION m6: Major difference in effect size

Lines 501-502 – what would be considered a “major” difference in effect size?

ANSWER m6: We consider a substantial difference when the coefficients doubled. The text now says page 7, lines 192-194: “We repeated the ExWAS in children only of European ancestry, and did not note substantial differences in effect size (more than doubling) between the two models (Figure 3A).”

As suggested by the first reviewer, we have moved this part of the manuscript, and have added Figure 3, to visualize the potential effect size difference as a scatter plot.

REVIEWER COMMENTS

Reviewer #2 (Remarks to the Author):

I appreciate the author's edits and replies. I believe they are comprehensive and addressed all of my major comments. Here are some very minor comments:

- Table S1B is erroneously labelled as Table S2
- Page 15 – lines 444-446 – what does the authors mean by “mainly”? As in, the highest number? If so, please consider quantifying that by saying “in our study design” (especially as it relates to the point earlier in the paragraph). For context, I am making this comment in part due to my original comment (Reviewer 2, M5).

Reviewer #3 (Remarks to the Author):

In this revised manuscript, the authors have described an encompassing undertaking, assessing a large number of environmental exposure and multiple biological measurements during pregnancy and childhood, in order to determine period-specific associations. I was not a reviewer in the first round, but after reading the manuscript and supplements I did review the response to reviewers, and commend both the original reviewers for their thoroughness as well as the authors for their detailed and thoughtful revision. Similar to a previous reviewer, I found accessing the HELIX catalog to require a password which as far as I could tell was not provided.

Overall while primarily exploratory, the breadth of these analysis combined with targeted sensitivity (BMI and ethnicity) and confounder (Cd and smoking) analysis make for an interesting story. While the total sample size is a bit concerning given the huge number of tests performed, and the limitation to primarily white participants might limit its extension to other populations, the analyses seem sound and the presentation of results is clear. I do have a few comments, divided into major, minor, and suggestions.

Major comments:

Confounding remains an issue here. The authors have been clear that it is a concern, but I would like to see something more systematic at least to be upfront about how correlated the environmental exposures are. The authors did an in-depth analysis of maternal smoking and Cd, but not cotinine, which should also be highly correlated, highlighting that there are many non-independent tests being performed here.

The cohort heterogeneity analysis seems truncated. A further analysis about which cohorts are outliers in which connections with an exploration about why would be valuable, as would a feature to the summary data in the catalog that shows the forest plots for each association to allow researchers to evaluate the association across cohorts.

Line 280, Figure 7C, in reference to the co-exposure between fish and pollutants. Some additional evidence to support this would be helpful (references or a description of the ExplosomeExplorer), right now it just looks like two associations.

The discussion of mechanisms and pathways (lines 404-442) does not include the very important caveat of tissue type. DNA methylation of blood demonstrating signatures for axon development and cognition does not actually mean that these signatures would also be present in the brain.

There is no mention at all of genetic effects, which would be extremely likely to have correlated effects on metabolites and DNAm, in particular.

Minor comments:

Figure 1 does not have sections, fFig 1 subsection references (Fig 1A, 1B, etc) do not make sense.

References to 3A and 3B are reversed in the text.

Suggestions, these are not required changes, just thoughts that struck me while reading:

Would we really expect the prenatal and postnatal top 10 exposures to be so different? The exposures measured were not identical but also not that different, what is the rationale about why the significant exposures and significant omics are so different prenatal and postnatal? The authors discuss the idea that prenatal exposure are more associated with DNAm as a more long term memory as opposed to more short term associations seen in childhood, but there is no real reason to assume that the pregnancy exposures were more distal to the measurement of DNAm in cord blood than the postnatal exposures were to the measurement of DNAm in child blood.

Related, only 5 exposures were associated with the same molecular features prenatally and postnatally. How correlated were these between prenatal and postnatal periods compared to the other exposures? Ie are they the most correlated ones or not?

Have you looked at the weather associations in terms of season? Seasonality affects diet and in particular allergen exposures. Also in terms of latitude – do the weather patterns have higher effect sizes in more northern regions with greater variation in weather?

Reviewer #4 (Remarks to the Author):

This is an interesting and novel manuscript that attempts to evaluate the environmental influence (part of the exposome) on multi-omics signatures, prenatally and in early-childhood. A large effort was made to include as many exposures as possible, but this does not entirely reflect the exposome. The manuscript provides a record or catalog of multiple associations without hypotheses. Important weaknesses of the current manuscript are lack of clear hypotheses, lack of adequate statistical modeling of multiple exposures/-omic, and no integration of multi-exposure or -omic layers. The manuscript is novel but difficult to follow since its descriptive nature reads more like a catalog. A major issue is that no attempt was made to model the multi exposure and multi -omic responses since they have layered data this would have been the most interesting findings. I have outline remaining issues that were not addressed below along with comments to improve interpretability:

QUESTION M4: Confounders

- There is substantial risk that the statistical models remain miss specified. Individual exposure/-omic statistical models might be miss specified since a minimal set of cofounders were used. For example, reviewer 1 highlighted the fact that Cd was not adjusted for smoking so we are likely seeing the effect of smoking that might or might not reflect Cd smoking.

- The minimal set of confounders: cohort, age, sex, z-score body mass index (z-BMI), ancestry, maternal education might not work for every -omic outcome and no conceptual model is presented for -omic outcomes where confounding might remain.

QUESTION M5: Methods of the ExWAS

- The authors test each exposure and -omic layer independently which defeats the purpose of referring to this as the exposome (i.e. one can do multiple studies of single chemicals and achieve similar results). The novelty and uniqueness of this study would be if the exposures were measured in the same individuals to look at interactions, effect modification and independent effects while adjusting for other exposures.

- No attempt was made to model multi exposure response function, interactions and effect modification that might more accurately reflect the human exposome with multi -omic layers. This would have more closely resembled the exposome impact on -omics.

QUESTION M8: Network analyses

- The network analysis might be bias given the individual testing of hypothesis (i.e. Cd and smoking signatures are very similar and therefore will bias the network).

QUESTION m1: Multivariate analyses

- The authors point at two papers one published and another under review where they evaluate the 1) expression vs DNA methylation and 2) SNPs -> urinary metabolites. The area of multi -omic integration is a miss opportunity in the current manuscript to look at shared -omic signatures influenced by multi exposures simultaneously and independently.

- The authors respond that the exposome will be explored in the future using multivariable approaches like CCA or PLS. Given the scope of the current framing on the exposome the multi-exposure response function needs to be explored in the current manuscript which will likely lead to major reworking of the manuscript.

REVIEWER COMMENTS

Reviewer #2 (Remarks to the Author):

I appreciate the author's edits and replies. I believe they are comprehensive and addressed all of my major comments.

We appreciate the comments from reviewer 2. They helped us to improve the manuscript.

Minor comments:

QUESTION m1: Table S1B - Table S2.

Table S1B is erroneously labelled as Table S2.

ANSWER m1: This has been changed in the updated version of the manuscript.

QUESTION m2: "Mainly captured by the methylome"

Page 15 – lines 444-446 – what do the authors mean by "mainly"? As in, the highest number? If so, please consider quantifying that by saying "in our study design" (especially as it relates to the point earlier in the paragraph). For context, I am making this comment in part due to my original comment (Reviewer 2, M5).

ANSWER m2: The original sentence was as follows: "Besides the specific exposure-omics associations, our study indicates that the choice of molecular layer and biological matrix is key in the design of exposome studies. We found that the pregnancy environment is mainly captured by the methylome with sustained changes until childhood. This goes in line with previous publications that suggest that the epigenome acts as the main source of cellular 'memory' and plasticity^{79,80}."

In order to address the comment, we have changed this to: "Our study indicates that the choice of molecular layer and biological matrix is key in the design of exposome studies. Most of the associations we found for the pregnancy exposome involved the methylome (70% of the associations observed). This is in line with previous publications that suggest that the epigenome acts as the main source of cellular 'memory' and plasticity^{82,83}. Although, it may partially reflect the nature of our study design and omics coverage (number of markers analysed in each omics layer and their intra-omics correlations)."

Reviewer #3 (Remarks to the Author):

In this revised manuscript, the authors have described an encompassing undertaking, assessing a large number of environmental exposure and multiple biological measurements during pregnancy and childhood, in order to determine period-specific associations. I was not a reviewer in the first round, but after reading the manuscript and supplements I did review the response to reviewers, and commend both the original reviewers for their thoroughness as well as the authors for their detailed and thoughtful revision. Similar to a previous reviewer, I found accessing the HELIX catalog to require a password which as far as I could tell was not provided.

We apologise for this. The user and password were provided through the Cover Letter. They are:
user: helix
pass: HELix888OmicS@

Overall, while primarily exploratory, the breadth of these analyses combined with targeted sensitivity (BMI and ethnicity) and confounder (Cd and smoking) analysis make for an interesting story. While the total sample size is a bit concerning given the huge number of tests performed, and the limitation to primarily white participants might limit its extension to other populations, the analyses seem sound and the presentation of results is clear. I do have a few comments, divided into major, minor, and suggestions.

We thank the reviewer for his/her positive feedback. Despite the limitations mentioned by the reviewer, we think the ExWAS catalogue will be a key tool in the exposomics field and will guide the design of future studies in a broader range of populations.

Major comments:

QUESTION M1: Confounding and multi-exposure models

Confounding remains an issue here. The authors have been clear that it is a concern, but I would like to see something more systematic at least to be upfront about how correlated the environmental exposures are. The authors did an in-depth analysis of maternal smoking and Cd, but not cotinine, which should also be highly correlated, highlighting that there are many non-independent tests being performed here.

ANSWER M1: Besides specific technical variables of each omics, models were adjusted for main confounding variables: cohort, child's age, sex, z-score body mass index (z-BMI), ancestry, maternal education, which captures several socio-economic variables. However, we agree with the reviewer that exposure variables might act as confounders among each other (i.e. fish consumption and Hg). To answer the reviewer's comment we provide the following additional analyses:

- A) Correlation matrices: We created a matrix of cohort adjusted correlations between all pregnancy exposures (Table S1B), and between all childhood exposures (Table S1C). In addition, the correlations adjusted for cohort between the same exposure across the two periods was already shown (now in Table S1C).
- B) Models adjusted for co-exposures: For the top 1,170 exposure-omics associations we now provide period-specific multi-exposure models for all omics biomarker that were associated to more than one "independent" exposure, where "independent" means exposures with a correlation <0.8 and belonging to a different mechanistic family (see Table S4). In these multi-exposure models, we fitted linear regression models between the omics biomarker and all independent exposures associated with that omics biomarker (i.e. correlation <0.8 , and not belonging to the same exposure group). These multi-exposure models were adjusted for the same covariates as the main model. Then, we estimated the percent change in effect size between the model with only one exposure and the multi-exposure model. Results obtained from these models are shown in Tables S4A-4B.

We revised the main text across the relevant sections to add this information, especially focusing on exposure-omics associations that were affected by the additional adjustment for co-exposures:

In the “Robustness of results with respect to ancestry, child zBMI and cohort” section (lines 212-221) we added: “Finally, given the correlated nature of the exposome, we ran multi-exposure models for those omics features associated with more than one exposure, when these exposures had a correlation <0.8 and belonged to different exposure families (except individual exposures that belonged to diet, metals or parabens that we considered as separate groups). Results of these analyses are shown in Tables S4A-4B. For prenatal exposures, the strongest effect change was observed for maternal cadmium (Cd) levels, which showed a reduction of >25% of the association with the molecular trait adjusting for smoking related variables. For childhood exposures, the strongest effect changes were observed for Hg, As, Se, PFAS and dietary patterns (e.g. fish and KIDMED score), for indoor PM and parental smoking, and for BPA and meteorological variables. They are discussed below in more detail.”

And sentences across the text:

- Lines 267-268 (regarding smoking and cadmium): “The multi-exposure analyses suggested some overlap between these signals (Table S4A).”
- Lines 282-283 (regarding postnatal air pollution and parental smoking): “Some of these associations were attenuated after adjusting for parental smoking (Table S4B)”.
- Lines 296-299: “In addition, multi-exposure models confirmed that most of these associations in particular with Hg, As and PFAS were attenuated after adjusting for diet and other co-exposures. This was not true for TMAO and As which remained one of the strongest association even after adjusting for PCB180, Hg, Fish and PFUNDA (Table S4B).”
- Lines 334-335 (regarding Cu): “Adjusting for co-exposures (e.g. Pb) did not substantially change these associations (Table S4B).”
- Lines 357-359: “The magnitude of some of these associations (carnitine C5, adiponectin, serotonin) were attenuated by more than 50% after adjusting for exposure to bisphenol A (BPA), which was previously found to reduce adiponectin release⁴⁷ (Table S4B).”
- Lines 417-428: “The strongest, most significant associations among all exposome-omics tested were found for As and Hg with trimethylamine-N-oxide (TMAO) and glycerophospholipids. Most of these associations were attenuated after adjusting for fish intake and other fish-related compounds. Indeed, TMAO was previously demonstrated to discriminate high against low fish intake, whereas homarine (a metabolite found in shellfish muscle) for high/non shellfish intake in populations with high seafood intake such as in Spain and Japan^{59,60}. TMAO-As association which remained the strongest association after adjusting for fish related exposures also suggests the independent role of the gut microbiome.”

QUESTION M2: Heterogeneity

The cohort heterogeneity analysis seems truncated. A further analysis about which cohorts are outliers in which connections with an exploration about why would be valuable, as would a feature to the summary data in the catalogue that shows the forest plots for each association to allow researchers to evaluate the association across cohorts.

ANSWER M2: We agree with the reviewer that heterogeneity among cohorts is an important point. Because of this, we are now presenting the forest plots of the 1,170 exposure-omics associations in Fig. S2 of the manuscript.

In addition, we have added a sentence in the Discussion about this issue: “Fifth, some associations presented high heterogeneity across cohorts. This can be explained by the different exposure levels, the different correlation with confounders, or the relatively small sample size within each cohort.”

QUESTION M3: Fish vs pollutants

Line 280, Figure 7C, in reference to the co-exposure between fish and pollutants. Some additional evidence to support this would be helpful (references or a description of the ExposomeExplorer), right now it just looks like two associations.

ANSWER M3: We have clarified in results how Exposome explorer was used by the following sentence: “Using systematic metabolite-diet associations found in previous population studies archived in ExposomeExplorer36, we confirmed the dietary origin of these exposure-metabolite associations, in this case to fish intake and animal products (Figure 7C).”

Moreover, we have moved some text related to this point from Results to the Discussion and have provided further evidence for fish-related associations from the literature: “Also, our study demonstrates the ability of metabolomics to accurately reflect dietary sources of exposures, corroborating previous literature on metabolite-diet associations. Some of the strongest associations were found for urinary biomarkers for seafood intake such as trimethylamine-N-oxide (TMAO) and taurine, respectively, which were previously demonstrated to discriminate high against low fish intake, whereas homarine (a metabolite found in shellfish muscle) for high/non shellfish intake in different populations in Spain and Japan. These metabolites were also previously found to be associated with Hg and As in pregnant women from the Spanish INMA cohort⁵⁹. Serum and urine metabolic signatures in our study suggest that children were exposed simultaneously through fish intake to essential elements and diverse bio-accumulated toxic compounds. Serum and urine metabolic signatures in our study suggest that children were exposed simultaneously through fish intake to essential elements and diverse bio-accumulated toxic compounds.”

QUESTION M4: Tissue-specific pathways

The discussion of mechanisms and pathways (lines 404-442) does not include the very important caveat of tissue type. DNA methylation of blood demonstrating signatures for axon development and cognition does not actually mean that these signatures would also be present in the brain.

ANSWER M4: We thank the reviewer for the comment. We have addressed it in the Discussion:

“Second, similarly, pathways identified for tobacco smoke (axon development, cognition, cholinergic synapse, insulin signalling, and several types of cancer) were in line with the effects of maternal smoking on health outcomes detected in HELIX children (higher blood pressure⁶⁵ and BMI⁶³, and increased behavioural problems⁶⁶). We acknowledge that, as DNA methylation was measured in blood, the identification of pathways relevant for other tissues (ie. Brain and axon development) has to be analysed with caution. It could be that DNA methylation marks are maintained across tissues if exposure happens early in development, or that the same genes are involved in different pathways in different tissues.”

QUESTION M5: Genetics

There is no mention at all of genetic effects, which would be extremely likely to have correlated effects on metabolites and DNAm, in particular.

ANSWER M5: We agree with the reviewer that genetic variation could also modify omics levels (metabolites, DNA methylation and also gene and miRNA expression and protein levels), alone or in combination with the exposures. However, this question was outside the scope of this manuscript. We have mentioned this limitation in the Discussion:

“Our study also has some limitations. First, omics platforms have a coverage bias and biological interpretability issues. For instance, the LC-MS/MS (Biocrates) method has a low coverage and does not give specific fatty acid side-chain composition for lipids, but it is widely used in large cohort studies and provides reproducible measurements with unambiguous annotation, easily comparable to other studies^{81–85}. We note that there are additional molecular layers and omics technologies of interest for future exposome studies, which were not included in our study, such as the gut metagenome, sensitive high-resolution mass spectrometry or single cell methods^{86–88}. Moreover, the effect of genetic variation, alone or in combination of the exposome, was not considered in this study.”

Minor comments:

QUESTION m1: Figure 1

Figure 1 does not have sections, Fig 1 subsection references (Fig 1A, 1B, etc) do not make sense.

ANSWER m1: Thank you for noting this. We have corrected it.

QUESTION m2: References to 3A and 3B are reversed in the text.

ANSWER m2: Thank you for noting this. We have updated the plot and the legend switching plots A and B.

Suggestions:

Suggestions, these are not required changes, just thoughts that struck me while reading:

SUGGESTION S1: Would we really expect the prenatal and postnatal top 10 exposures to be so different? The exposures measured were not identical but also not that different, what is the rationale about why the significant exposures and significant omics are so different prenatal and postnatal? The authors discuss the idea that prenatal exposure are more associated with DNAm as a more long term memory as opposed to more short term associations seen in childhood, but there is no real reason to assume that the pregnancy exposures were more distal to the measurement of DNAm in cord blood than the postnatal exposures were to the measurement of DNAm in child blood. Related, only 5 exposures were associated with the same molecular features prenatally and postnatally. How correlated were these between prenatal and postnatal periods compared to the other exposures? Are they the most correlated ones or not?

ANSWER S1: We have analysed the pregnancy and childhood exposome vs child molecular profiles. Unfortunately, we have not been able to analyse molecular profiles at birth. Thus, when we find an association with the pregnancy exposome this implies a persistent effect (exposure in pregnancy - effect still seen in childhood). It also could be that the pregnancy association reflects the effect of

the childhood exposure, if inter-period correlation of the exposure is high. In this situation we should see associations with both periods.

Some potential explanations for the low overlap of exposure-omics associations between exposure periods are:

- 1) Low inter-period correlation between the exposure levels (see Table S1D with all the pregnancy-childhood correlations for the same exposure).
- 2) Even if correlation is high, the same exposure might have different mechanisms of action (different route, dose, etc.) in the mother (pregnancy period) and in the child. A good example of this is the exposure to tobacco smoke, which refers to maternal active smoking during the pregnancy period and to exposure to second-hand smoke in the childhood exposome.
- 3) The dynamics of the molecular response (acute vs. persistent), also can explain the low overlap. For instance, in our study, acute effects are only captured for the childhood exposome, but not for the pregnancy exposome (where the outcome is assessed 8 years later).

To address this comment, we have added the following sentence in the Discussion section: “Finally, we observed little overlap in associations for the pregnancy and childhood exposome, likely due to the low inter-period correlation of exposures, the differences in the exposure route or dose between periods, and the dynamics of the molecular response (ie. our study is able to capture long-term responses of the pregnancy exposome but only short-term responses of the childhood exposome)”

SUGGESTION S2: Weather and season-latitude

Have you looked at the weather associations in terms of season? Seasonality affects diet and in particular allergen exposures. Also in terms of latitude – do the weather patterns have higher effect sizes in more northern regions with greater variation in weather?

ANSWER S2: We have not explicitly looked at season or latitude as exposure variables.

Regarding the former, all cohorts collected biological samples across seasons, except for EDEN (Poitiers, France) that did not visit children during autumn (Figure 1). Regarding the later, the latitudes of the HELIX cohorts are: MoBA (Oslo - 59.9), KAUNAS (Kaunas - 54.9), BiB (Bradford - 53.8), EDEN (Poitiers - 46.3), INMA (Sabadell - 41.5), Rhea (Heraklion - 35.4).

Figure 1. Number of children visited per month (1-12) in each cohort.

Given that weather conditions depend on latitude, thus on cohort, we decided not to analyse average annual levels, which would be the same as analysing cohort effects, but average levels of the month before biological sample collection. As the reviewer suggests, monthly average weather conditions can be capturing factors related to seasonal variation. Indeed, in the Discussion we already mentioned this: “Sixth, our results also provide insights into potential mechanisms of action for weather conditions: they appear to have direct effects (e.g. regulating thermogenesis) and indirect effects (e.g. determining other exposures such as virus survival), or they can also represent proxies of other variables (e.g. hours of daylight). The investigation of meteorological conditions in larger longitudinal omics datasets covering seasonal patterns will be needed to elucidate the final causal mechanisms.” We have added season as another potential explanation: “or they can also represent proxies of other variables (e.g. hours of daylight or dietary changes due to seasonal variation).”

Reviewer #4 (Remarks to the Author):

This is an interesting and novel manuscript that attempts to evaluate the environmental influence (part of the exposome) on multi-omics signatures, prenatally and in early-childhood. A large effort was made to include as many exposures as possible, but this does not entirely reflect the exposome. The manuscript provides a record or catalog of multiple associations without hypotheses. Important weaknesses of the current manuscript are lack of clear hypotheses, lack of adequate statistical modelling of multiple exposures/-omic, and no integration of multi-exposure or -omic layers. The manuscript is novel but difficult to follow since its descriptive nature reads more like a catalog. A

major issue is that no attempt was made to model the multi exposure and multi -omic responses since they have layered data this would have been the most interesting findings. I have outline remaining issues that were not addressed below along with comments to improve interpretability:

Before answering in detail the different questions raised by reviewer 4, we would like to highlight that, although our study does not cover the whole exposome (which is impossible to be measured by nature), it evaluates almost 200 exposure variables covering very different environment domains (chemical, urban, social, diet, etc..) representing a huge advance over previous studies that only evaluated one or a few exposures or one exposure family.

QUESTION M1: Confounders

There is substantial risk that the statistical models remain miss specified. Individual exposure/-omic statistical models might be miss specified since a minimal set of cofounders were used. For example, reviewer 1 highlighted the fact that Cd was not adjusted for smoking so we are likely seeing the effect of smoking that might or might not reflect Cd smoking. The minimal set of confounders: cohort, age, sex, z-score body mass index (z-BMI), ancestry, maternal education might not work for every -omic outcome and no conceptual model is presented for -omic outcomes where confounding might remain.

Besides specific technical variables of each omics, models were adjusted for main confounding variables: cohort, child's age, sex, z-score body mass index (z-BMI), ancestry, maternal education, which captures several socio-economic variables. However, we agree with the reviewer that exposure variables might act as confounders between each other (e.g. fish consumption and Hg). Indeed, this was already recognized in the Discussion: "Finally, although our models were adjusted for confounders, residual confounding might still be present and causal links would need to be proven through interventions, Mendelian randomization analyses, cross-contextual studies, or in vivo / in vitro models in order to move to therapeutic and preventative strategies."

In any case, we see the concern by the reviewer and have tried to answer this question together with QUESTION M3: Multi-exposure models.

QUESTION M2: ExWAS

The authors test each exposure and -omic layer independently which defeats the purpose of referring to this as the exposome (i.e. one can do multiple studies of single chemicals and achieve similar results). The novelty and uniqueness of this study would be if the exposures were measured in the same individuals to look at interactions, effect modification and independent effects while adjusting for other exposures.

ANSWER M2: The manuscript was designed as a hypothesis free (or screening) exposome-omics-wide association study (ExWAS), where each exposure was associated with each molecular marker, similar as it is done in genome-wide association studies (GWAS). This allows us to provide a first catalogue of exposome-omics associations that other studies can build on.

However, we understand the concern about confounding between exposures (multi-exposure models). See below our answer to this point. Regarding interactions between exposures we think the study does not have the statistical power to address this.

QUESTION M3: Multi-exposure models

No attempt was made to model multi exposure response function, interactions and effect modification that might more accurately reflect the human exposome with multi -omic layers. This would have more closely resembled the exposome impact on -omics.

ANSWER M3: Following the advice of this reviewer and reviewer #3, we have run new multi-exposure models, see answer M1.

We agree with the reviewer that we did not systematically explore the multi-exposure interactions and effect modifications, except for a few cases. This was mentioned in the Discussion: “Moreover, the correlated nature of the exposome makes identification of driving exposures difficult. Here we tried to separate the effect of maternal tobacco smoking and maternal cadmium (Cd) by running stratified models and identified Cd-specific effects. Besides tobacco smoke, Cd might have other origins such as rice, potatoes and wheat, when frequently consumed in large quantities⁹¹. Mixture or multi-pollutant approaches aim to tackle this more systematically, however these are not yet suitable for high-dimensional omics datasets such as ours^{92,93}.”.

QUESTION M4: Network analyses

The network analysis might be biased given the individual testing of hypothesis (i.e. Cd and smoking signatures are very similar and therefore will bias the network).

It is true that networks capture the correlation of the exposures and the correlation of the molecular markers. This was already mentioned in the manuscript: “This connectivity highlights the correlated nature of the serum and urine metabolome, which represented the majority of the exposure-omics associations of the network (43 and 26% respectively)”. However, we still think networks have an important value in allowing the visualisation of all correlations and simplification of the ExWAS findings.

Recognising the concerns of the reviewer, we added additional information in the Discussion: “By partitioning these associations into network clusters for visualisation and by conducting systematic biological interpretation, this study reveals potential biological responses and sources of exposure.”.

QUESTION M5: Multivariate analyses

The authors point at two papers one published and another under review where they evaluate the 1) expression vs DNA methylation and 2) SNPs -> urinary metabolites. The area of multi-omic integration is a missed opportunity in the current manuscript to look at shared -omic signatures influenced by multi exposures simultaneously and independently.

The authors respond that the exposome will be explored in the future using multivariable approaches like CCA or PLS. Given the scope of the current framing on the exposome the multi-exposure response function needs to be explored in the current manuscript which will likely lead to major reworking of the manuscript.

We thank the reviewer for the recommendation of presenting results of multi-omic/multivariate models in this manuscript, however we would like to emphasize the limitations of these agnostic approaches in the frame of our research question.

Our scientific question was to decipher which omic features are associated with exposures in order to build a catalogue, for which the most appropriate method is an ExWAS. An alternative analytical

approach could have been to use multivariate latent variable approaches, such as PLS or CCA, to model the covariance within and between omics layers agnostically, capturing functional relationships that naturally occur, prior to association to exposures. Multivariate methods are more oriented towards performing descriptive analyses. They would allow describing clusters of Exposures and omic features which are correlated among them but do not allow to obtain significant threshold with p-values and the criteria are based of selecting some principal components. These methods do not readily allow for confounder adjustment and results are highly sensitive to adjustable parameters in implementation, for example the precise objective function to be optimised, the relative scaling of variables and the number of latent variables selected. In addition, the large omics, such as methylation and transcriptomics, require prior data reduction before applying this type of model, which means potential loss of information.

Therefore, for the present study we opted for an analysis which was driven by the exposome and maximised true signal recovery by appropriate adjustments. ExWAS provide easy biological interpretation and comparison with other studies (ie. EWAS catalogue and EWAS atlas).

Integration of the “omics” layers was included *posteriori* and biologically driven (instead of agnostically/data driven), focusing on those omics markers associated with the exposures, through cross-omics pathway enrichment analyses. In addition, replication of exposure-omics associations across molecular layers and biological matrices was checked in detail in the last paragraph of results.

Just to clarify the two papers mentioned by the reviewer, 1) expression vs DNA methylation and 2) SNPs vs urinary metabolites, do not include exposures, just omics correlation.

REVIEWERS' COMMENTS

Reviewer #3 (Remarks to the Author):

I appreciate the time the authors took to address my comments and suggestions, very nice work!

Reviewer #4 (Remarks to the Author):

The authors have addressed some concerns regarding the multi-exposure models but not multi -omic signatures with the exposome. I think the catalog of evidence from early-life exposures is worth presenting.

A minor comment is to rephrase the title from "early life exposome" to "early life exposures" since the authors acknowledge that the exposures are limited.

ANSWERS TO REVIEWERS:

Reviewer #3 (Remarks to the Author):

I appreciate the time the authors took to address my comments and suggestions, very nice work!

We thank the reviewer for his/her contributions to the development of this manuscript.

Reviewer #4 (Remarks to the Author):

The authors have addressed some concerns regarding the multi-exposure models but not multi-omic signatures with the exposome. I think the catalog of evidence from early-life exposures is worth presenting.

We thank the reviewer for his/her appreciation of the value of the comprehensive and systematic catalogue of all exposure-omics associations. As explained previously, the univariate (ExWAS) approach is indeed the most appropriate for this.

A minor comment is to rephrase the title from "early life exposome" to "early life exposures" since the authors acknowledge that the exposures are limited.

We acknowledge that our exposome does not reflect the whole exposome. However, it is well recognized that the entire exposome is impossible to measure by its nature, and that a partial characterization is the only possible way to implement exposome studies. Indeed, other studies focus on much smaller parts of the exposome (e.g. only chemical exposome, or only urban exposome). We evaluate almost 200 exposure variables covering very different environment domains (chemical, urban, social, diet, ...) representing a huge advance over previous studies that only evaluated one or a few exposures or one exposure family in relation to omics markers. Furthermore, we assess these exposures in two critical developmental time periods, including highly sensitive biomarkers for many chemical exposures and wide-ranging geospatial modelling of the outdoor and built environment.

For these reasons, and because only one of the four reviewers proposed this change our title as a minor comment, we have decided not to make this change.